# Dendritic *atoh1a+* cells serve as Merkel cell precursors during skin development and regeneration

Evan W. Craig[1], Erik C. Black[1,2], Samantha Z. Fernandes[1,2], Ahlan S. Ferdous[1], Camille E. A. Goo[1], Sheridan M. Sargent[1,3], Elgene J. A. Quitevis[1], Avery Angell Swearer[1,2], Nathaniel G. Yee[1], Jimann Shin[4], Lilianna Solnica-Krezel[4] and Jeffrey P. Rasmussen[1,5,*]

## ABSTRACT

Sensory cells often adopt specific morphologies that aid in the detection of external stimuli. Merkel cells encode gentle touch stimuli in vertebrate skin and adopt a reproducible shape characterized by spiky actin-rich microvilli that emanate from the cell surface. The mechanisms by which Merkel cells acquire this stereotyped morphology from keratinocyte progenitors are unknown. Here, we establish that dendritic Merkel cells (dMCs) express *atonal homolog 1a* (*atoh1a*), extend dynamic filopodial processes, and arise in transient waves during zebrafish skin development and regeneration. We find that dMCs share molecular similarities with both basal keratinocytes and Merkel cells, yet display mesenchymal-like behaviors, including local cell motility and proliferation within the epidermis. Furthermore, dMCs can directly adopt the mature, microvilliated Merkel cell morphology through substantial remodeling of the actin cytoskeleton. Loss of Ectodysplasin A signaling alters the morphology of dMCs and Merkel cells within specific skin regions. Our results show that dMCs represent an intermediate state in the Merkel cell maturation program and identify Ectodysplasin A signaling as a key regulator of Merkel cell morphology.

KEY WORDS: Ectodysplasin, Epidermis, Cell motility, Somatosensory system, Zebrafish, Piezo2, Tp63, Microvilli

## INTRODUCTION

Organ development and function require that constituent cells adopt precise shapes. For example, epithelial cells often develop actin-based membrane protrusions integral to organ function (reviewed by Sharkova et al., 2023). Defects in the morphogenesis or maintenance of actin-based protrusions are linked to several diseases (reviewed by Houdusse and Titus, 2021). Thus, elucidating how cells adopt

specific actin-based shapes is relevant to understanding both organ function and human pathologies.

Merkel cells (MCs) are mechanosensory epidermal cells that interact with somatosensory neurites to form the MC-neurite complex, which mediates gentle touch detection (reviewed by Woo et al., 2015). MCs display a remarkably consistent morphology across diverse vertebrate skin types (reviewed by Hartschuh et al., 1986; Whitear, 1989). The core morphological features of MCs include a small and spherical cell body, a high nuclear-to-cytoplasmic ratio and neurosecretory granules (Hartschuh et al., 1986; Whitear, 1989). Strikingly, MCs extend numerous actin-rich microvilli from the cell surface, giving MCs a 'mace-like' morphology (Hartschuh et al., 1986; Lane and Whitear, 1977; Sekerková et al., 2004; Takahashi-Iwanaga, 2003; Takahashi-Iwanaga and Abe, 2001; Toyoshima et al., 1998; Whitear and Lane, 1981). Lineage tracing indicates that MCs derive from basal keratinocyte precursors in mammalian and zebrafish skin (Brown et al., 2023; Morrison et al., 2009; Van Keymeulen et al., 2009), yet how cells within the MC lineage lose basal keratinocyte characteristics and adopt the hallmark MC morphology remains unknown. A deeper understanding of the MC lineage may inform studies of Merkel cell carcinoma, an aggressive skin cancer of unclear cellular origin (reviewed by Becker et al., 2017; Harms et al., 2018).

We recently established zebrafish as an *in vivo* model for MC studies (Brown et al., 2023). Here, we leverage the optical accessibility of zebrafish skin to directly observe cell behaviors during MC maturation. By visualizing a filamentous actin (F-actin) reporter expressed in MCs, we describe a morphologically distinct population of epidermal cells, termed dendritic Merkel cells (dMCs), that shares characteristics with keratinocytes and MCs. Importantly, we document the direct maturation of dMCs into MCs through cytoskeletal rearrangements. Furthermore, we show that Ectodysplasin A (Eda) signaling is required for MC morphology within trunk skin. Together, our results provide *in vivo* characterizations of MC precursor states and identify Eda signaling as a key regulator of zebrafish MC morphogenesis.

## RESULTS

### A transient and morphologically distinct population of keratinocyte-derived *atoh1a+* cells emerges during skin development

Staging of zebrafish post-embryonic development relies on standard length (SL) in millimeters, with sexual maturity attained ~18 mm SL (Parichy et al., 2009). Cells with the prototypical MC morphology – spherical with actin-rich microvillar protrusions of ~1-2 μm in length – populate several adult zebrafish skin compartments, including trunk skin (Fig. 1A,B; Brown et al., 2023). Atonal homolog 1 (*Atoh1*) encodes a transcription factor that is necessary and sufficient for

[1]Department of Biology, University of Washington, Seattle, WA 98195, USA. [2]Molecular and Cellular Biology Program, University of Washington, Seattle, WA 98195, USA. [3]Graduate Program in Neuroscience, University of Washington, Seattle, WA 98195, USA. [4]Department of Developmental Biology, Washington University School of Medicine, St Louis, MO 63110, USA. [5]Institute for Stem Cell and Regenerative Medicine, University of Washington, Seattle, WA 98109, USA.

*Author for correspondence (rasmuss@uw.edu)

E.C.B., 0000-0002-2333-8923; S.Z.F., 0000-0003-3687-5231; C.E.A.G., 0000-0002-9118-4006; J.S., 0000-0003-1027-0517; L.S., 0000-0003-0983-221X; J.P.R., 0000-0001-6997-3773

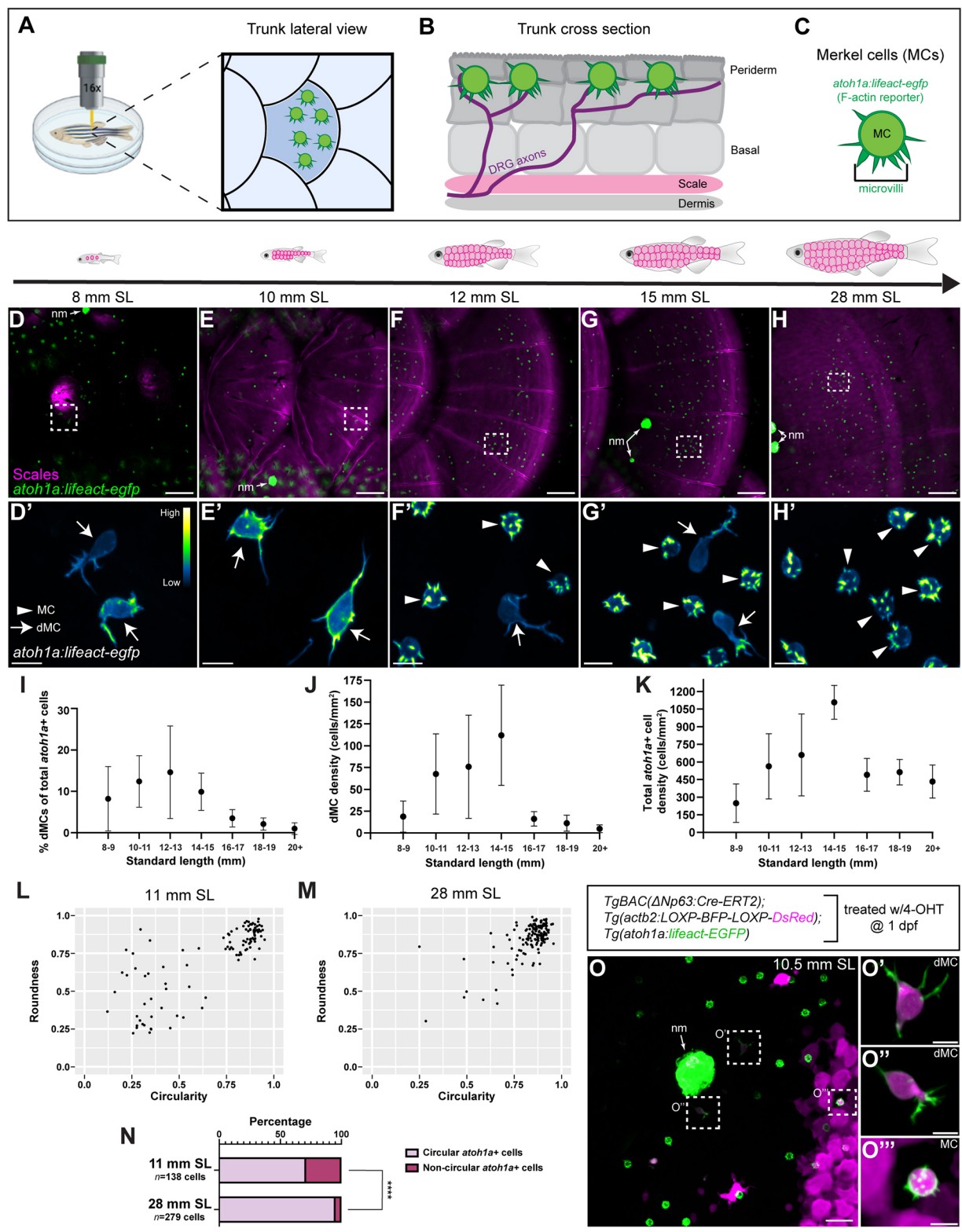

**Fig. 1.** See next page for legend.

murine MC development (Morrison et al., 2009; Ostrowski et al., 2015; Van Keymeulen et al., 2009). Zebrafish *atoh1a* is an ortholog of murine *Atoh1*, and zebrafish MCs express reporters inserted into the endogenous *atoh1a* upstream region (Brown et al., 2023). Using a nuclear-localized *atoh1a* reporter to label MCs, we have previously found that trunk MCs appear during the onset of squamation (scale

formation) at ~9 mm SL and increase in density from 10-15 mm SL (Brown et al., 2023). As the nuclear reporter did not allow the visualization of cell shapes, the morphology of developing zebrafish MCs remains unclear.

To assess MC morphology during squamation, we conducted a developmental staging time course using *Tg(atoh1a:lifeact-egfp)*,

**Fig. 1. Identification of a transient and morphologically distinct population of keratinocyte-derived *atoh1a+* cells during skin development.** (A) Schematic of live-imaging methodology for visualizing MCs (green) along the lateral zebrafish trunk. (B) Illustration of adult trunk skin in cross-section. Stratified layers of keratinocytes with interspersed MCs (green) reside above the bony scale (pink). (C) *Tg(atoh1a:lifeact-egfp)* allows visualization of the MC F-actin cytoskeleton, including microvilli. (D-H) Representative images of *atoh1a+* MCs [green; *Tg(atoh1a:lifeact-egfp)*] and scales (magenta; Alizarin Red S staining) along the lateral trunk at the indicated stages. Regions outlined in D-H are shown at higher magnification in D′-H′. (D′-H′) *Tg(atoh1a:lifeact-egfp)* signal intensity color-coded using the 'Green Fire Blue' lookup table. Arrowheads indicate MCs; arrows indicate dMCs. (I) Percentage of *atoh1a+* cells with dMC morphology out of total number of *atoh1a+* cells relative to SL. Each dot represents the mean dMC percentage calculated from $n\geq3$ animals. For each animal, multiple images were acquired and grouped to calculate the dMC percentage. (J) Plot of dMC density relative to SL from the dataset in I. (K) Plot of total *atoh1a+* cell density (dMCs and MCs) relative to SL from the dataset in I. Data are mean±s.d. (L,M) *atoh1a+* cell shape analysis for animals of the indicated stages (*n*=100 randomly selected cells displayed from one animal for each stage). Dots represent individual *atoh1a+* cells. Top right quadrant indicates cells with the most round and circular morphologies, indicative of MCs. Cells falling outside of the top right quadrant possess less round and circular morphologies, indicative of dMCs. (N) Stacked bar charts of *atoh1a+* cell shape analysis performed in L and M. *atoh1a+* cells in adult skin have significantly more circular morphologies than *atoh1a+* cells in juvenile skin (****$P<0.0001$; Fisher's exact test). (O-O‴) Confocal images of the trunk epidermis from a *TgBAC(ΔNp63:Cre-ERT2); Tg(actb2:LOXP-BFP-LOXP-DsRed); Tg(atoh1a:lifeact-egfp)* juvenile treated with 4-OHT at 1 dpf. There is mosaic DsRed expression (magenta) in basal keratinocytes and derivatives. DsRed+ dMCs shown in O′ and O″, along with a DsRed+ MC in O‴, indicate that both are derived from basal keratinocytes. nm, neuromasts containing clusters of *atoh1a+* hair cells. Scale bars: 50 μm in D-H; 20 μm in O; 5 μm in D′-H′,O′-O‴.

which expresses a F-actin reporter in MCs, allowing visualization of the MC actin cytoskeleton *in vivo* (Fig. 1C; Brown et al., 2023). To simultaneously visualize scales, we stained animals with Alizarin Red S, which labels the calcified scale matrix. Like our previous observations in adults, we identified cells with the prototypical MC morphology at late juvenile and adult stages (Fig. 1F-H′, arrowheads). Interestingly, we also observed *atoh1a+* cells with weaker transgene signal and highly variable morphologies, characterized by ovoid cell bodies and long filopodial-like actin-rich protrusions (Fig. 1D-G′, arrows). Previous studies described cells, which were referred to as dMCs, with similar oblong or dendritic morphologies labeled by immunostaining for cytokeratins 8, 18 and 20 in developing human plantar and hairy skin, in human and rodent oral mucosa, and in murine touch domes (Kim and Holbrook, 1995; Moll et al., 1984; Nakafusa et al., 2006; Narisawa et al., 1993; Tachibana et al., 1998, 1997). For consistency with the literature, we hereafter refer to these cells as dMCs, although we note that whether dMCs relate to MCs or represent an alternative cell fate has not been established. Manual cell counting from our imaging dataset revealed dMCs appeared at their highest frequency and density at 10-15 mm SL (Fig. 1I,J). During this period, scales expanded and total *atoh1a+* cell density continuously increased (Fig. 1K), consistent with our previous work (Brown et al., 2023). Next, we quantified *atoh1a+* cell shapes at representative juvenile and adult stages (11 and 28 mm SL, respectively) by assigning circularity and roundness values to thresholded images from our dataset. For this dataset, we defined 'circular' cells as those having circularity and roundness values >0.7. We found that 95.3% of *atoh1a+* cells in adult skin fell into the circular category, likely representing the mature microvilliated MC (Fig. 1M,N). By contrast, 71.1% of *atoh1a+* cells in juvenile skin fell into the circular category, with numerous *atoh1a+*

cells adopting more oblong shapes (Fig. 1L,N). Thus, our results indicate that a morphologically distinct population of *atoh1a+* cells transiently populates the trunk epidermis during squamation.

We previously found that most adult trunk MCs contact somatosensory axons (Brown et al., 2023). To determine whether dMCs also contacted somatosensory axons, we crossed *Tg(atoh1a:lifeact-egfp)* to *Tg(p2rx3a:mCherry)*, a reporter that labels a subset of cutaneous somatosensory axons (Palanca et al., 2013; Rasmussen et al., 2018), and acquired confocal *z*-stacks of juvenile skin from double transgenic fish (Fig. S1A). We found that 90.5% of dMCs contacted *p2rx3a+* axons (*n*=38/42 cells; Fig. S1A′,B) and that axon contacts could occur at the cell body or on a protrusion, or both (Fig. S1C). Thus, like MCs, dMCs frequently associate with cutaneous axons.

Using a tamoxifen-inducible Cre driver expressed in basal keratinocytes [*TgBAC(ΔNp63:Cre-ERT2)*; Brown et al., 2023] and a quasi-ubiquitous Cre reporter transgene [*Tg(actb2:LOXP-BFP-LOXP-DsRed)*; Kobayashi et al., 2014], we previously showed that zebrafish MCs are derived from *ΔNp63*-expressing embryonic basal keratinocytes (Brown et al., 2023). To determine whether basal keratinocytes give rise to dMCs, we treated *TgBAC(ΔNp63:Cre-ERT2); Tg(actb2:LOXP-BFP-LOXP-DsRed); Tg(atoh1a:lifeact-egfp)* embryos with *trans*-4-OH-tamoxifen (4-OHT) to induce Cre-ERT2 activity at 1 day post-fertilization (dpf). This resulted in permanent DsRed expression in basal keratinocytes and their derivatives. We then raised animals to squamation stages. Owing to transgene mosaicism or incomplete Cre-ERT2 activation, not all basal keratinocytes expressed DsRed (Fig. 1O). By thresholding the DsRed or BFP channel from maximum intensity projections, we found that 39.6% of the epidermis was DsRed+, whereas 35.2% was BFP+ (*n*=71 scales from 4 fish; overall recombination efficiency: 52.9%). Along with DsRed+ MCs (Fig. 1O‴), we observed that a subset of dMCs were DsRed+ (Fig. 1O′,O″). Although only 14.5% (*n*=48/332 cells) of dMCs expressed *Tg(actb2:LOXP-BFP-LOXP-DsRed)*, of these, 54.2% (*n*=26/48 cells) were DsRed+, consistent with the overall rate of epidermal recombination. Interestingly, we often identified DsRed+ dMCs at a distance from the nearest DsRed+ basal keratinocyte (Fig. 1O). These findings support a basal keratinocyte origin for dMCs.

## dMCs are the primary *atoh1a+* cell type during early stages of skin regeneration

Our identification of dMCs during squamation led us to posit that dMCs may also appear during scale regeneration. To induce scale regeneration, we performed scale 'plucking' with forceps to remove both the bony scale and overlying epidermis containing MCs (Fig. 2A). Scale removal triggers a wound healing response, causing neighboring keratinocytes to migrate and cover the denuded area within hours (Richardson et al., 2016; Santoso et al., 2024). Over several days, keratinocytes then proliferate and re-establish a stratified epidermis (Richardson et al., 2016), while dermal osteoblasts proliferate and undergo hypertrophy to regenerate the bony scale (Cox et al., 2018; De Simone et al., 2021; Iwasaki et al., 2018).

To determine whether MCs populated the regenerating scale epidermis, we plucked scales from animals expressing *Tg(atoh1a:Lifeact-egfp)* and *Tg(sp7:mCherry)* (Singh et al., 2012), an osteoblast reporter (Fig. 2A,B). As expected, scales underwent substantial regeneration within 7 days post-pluck (dpp) (Fig. 2B-E). *atoh1a+* cells appeared above regenerating scales beginning at 2 dpp and increased in density until 14 dpp (Fig. 2F-J). Strikingly, dMCs comprised ~90% of *atoh1a+* cells present at 2 dpp (Fig. 2G,G′,K)

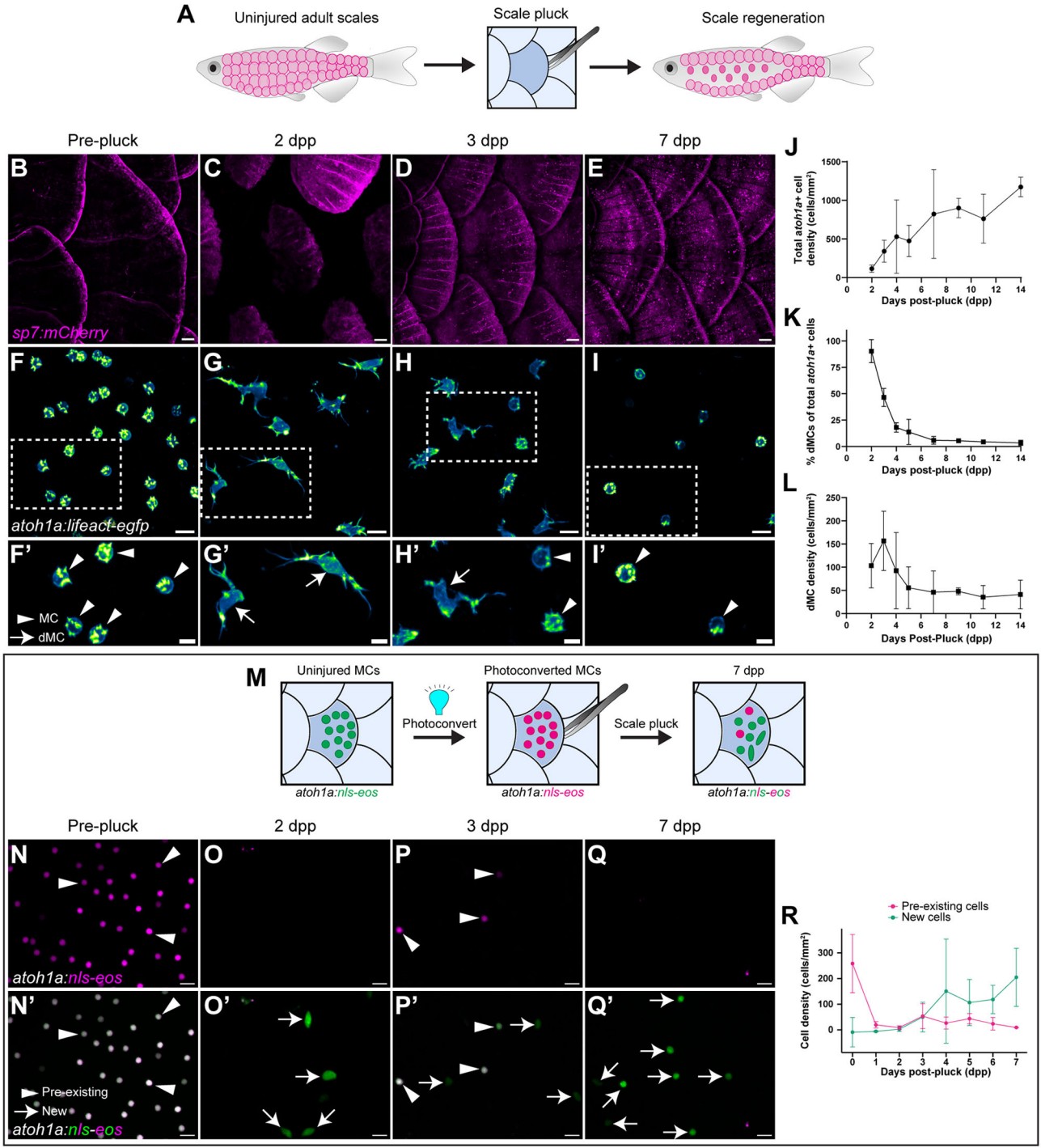

**Fig. 2. dMCs are the predominant *atoh1a*+ cell morphology during the early stages of skin regeneration.** (A) Illustration of the scale pluck regeneration model. Physical plucking triggers regeneration of dermal scales and overlying epidermis. (B-I′) Representative images of scale-forming osteoblasts [magenta; *Tg(sp7:mCherry)*] (B-E) or *atoh1a*+ cells within the scale epidermis (F-I) at the indicated stages. Arrowheads indicate MCs; arrows indicate dMCs. Areas outlined in F-I are shown at higher magnification in F′-I′. (J) Plot of total *atoh1a*+ cell density (dMCs and MCs) throughout scale regeneration. Each dot represents a result from confocal images collected from multiple zebrafish of the corresponding timepoint (2-7 dpp, *n*=8-13 fish; 9-14 dpp, *n*=2-4 fish). Total cells analyzed: 5764 MCs and 1064 dMCs. Data are mean±s.d. (K) Quantification of dMC frequency during scale regeneration from the dataset in J. Each dot represents the mean dMC frequency±s.d. (L) Plot of dMC density throughout scale regeneration from the dataset in J. Data are mean±s.d. (M) Illustration depicting the *atoh1a*+ cell photoconversion paradigm. *Tg(atoh1a:nls-eos)* expresses nuclear-localized Eos in dMCs and MCs. UV light exposure irreversibly photoconverts *atoh1a*+ cells in uninjured scales. Scale plucking then induces regeneration. Pre-existing *atoh1a*+ cells contain photoconverted nls-Eos (magenta) in the new scale region, whereas new *atoh1a*+ cells contain only non-photoconverted nls-Eos (green). (N-Q′) Representative images of the photoconverted *Tg(atoh1a:nls-eos)* scale epidermis pre-scale pluck (N,N′) and post-pluck (O-Q′). Single channel images of the photoconverted nls-Eos channel (magenta) are shown in N-Q. Merged images of photoconverted (magenta) and non-photoconverted nls-Eos (green) are shown in N′-Q′. Arrowheads indicate pre-existing cells (containing photoconverted nls-Eos); arrows indicate *de novo* generated cells (containing only non-photoconverted nls-Eos). (R) Quantification of pre-existing and new *atoh1a*+ cells at the indicated stages. Each dot represents the mean *atoh1a*+ cell density from *n*=3 or 4 fish. Data are mean±s.d. Scale bars: 100 µm in B-E; 10 µm in F-I; 5 µm in F′-I′; 10 µm in (N-Q′).

and ~45% of *atoh1a+* cells at 3 dpp (Fig. 2H,K). After 3 dpp, the proportion and density of dMCs gradually decreased, with MCs becoming the predominant *atoh1a+* cell type at later stages of regeneration (Fig. 2I,K,L).

dMCs and MCs on regenerating scales could arise from either *de novo* production or movement of pre-existing *atoh1a+* cells from surrounding, uninjured regions of epidermis. *De novo* production may include differentiation from precursors and/or trans-differentiation of another cell type. To distinguish between *de novo* production and movement of pre-existing cells, we irreversibly photoconverted *atoh1a+* cells expressing nuclear localized Eos [*Tg(atoh1a:nls-eos);* (Pickett et al., 2018)] and plucked scales (Fig. 2M,N). In this paradigm, pre-existing *atoh1a+* cells contain both photoconverted and non-photoconverted nls-Eos, whereas *atoh1a+* cells produced *de novo* contain only non-photoconverted nls-Eos. We previously established that photoconverted nls-Eos is stable in adult MCs for ≥1 month (Brown et al., 2023). Notably, at 2 and 3 dpp, the epidermis contained many *de novo* produced cells

with ovoid nuclei that likely represented dMCs (Fig. 2O,P,R). By 7 dpp, nearly all *atoh1a+* cells were produced *de novo* (Fig. 2Q,R). Based on these results, we concluded that skin injury triggers *de novo* production of dMCs and MCs within the trunk epidermis, and that early stages of epidermal regeneration are associated with a high proportion of dMCs. Thus, dMCs are a transient *atoh1a+* cell type during both skin development and regeneration.

## dMCs exhibit molecular features of both keratinocytes and MCs

To characterize the molecular properties of dMCs associated with regenerating scales, we focused on 5 dpp, a timepoint at which scales withstand immunostaining and contain a mix of dMCs and MCs (Fig. 2K). We first assayed expression of Tp63, a transcription factor that is crucial for epidermal development and enriched in basal keratinocytes (Bakkers et al., 2002; Guzman et al., 2013; Lee and Kimelman, 2002; Rangel-Huerta et al., 2021) (Fig. 3B′). Immunostaining revealed that MCs did not express detectable levels

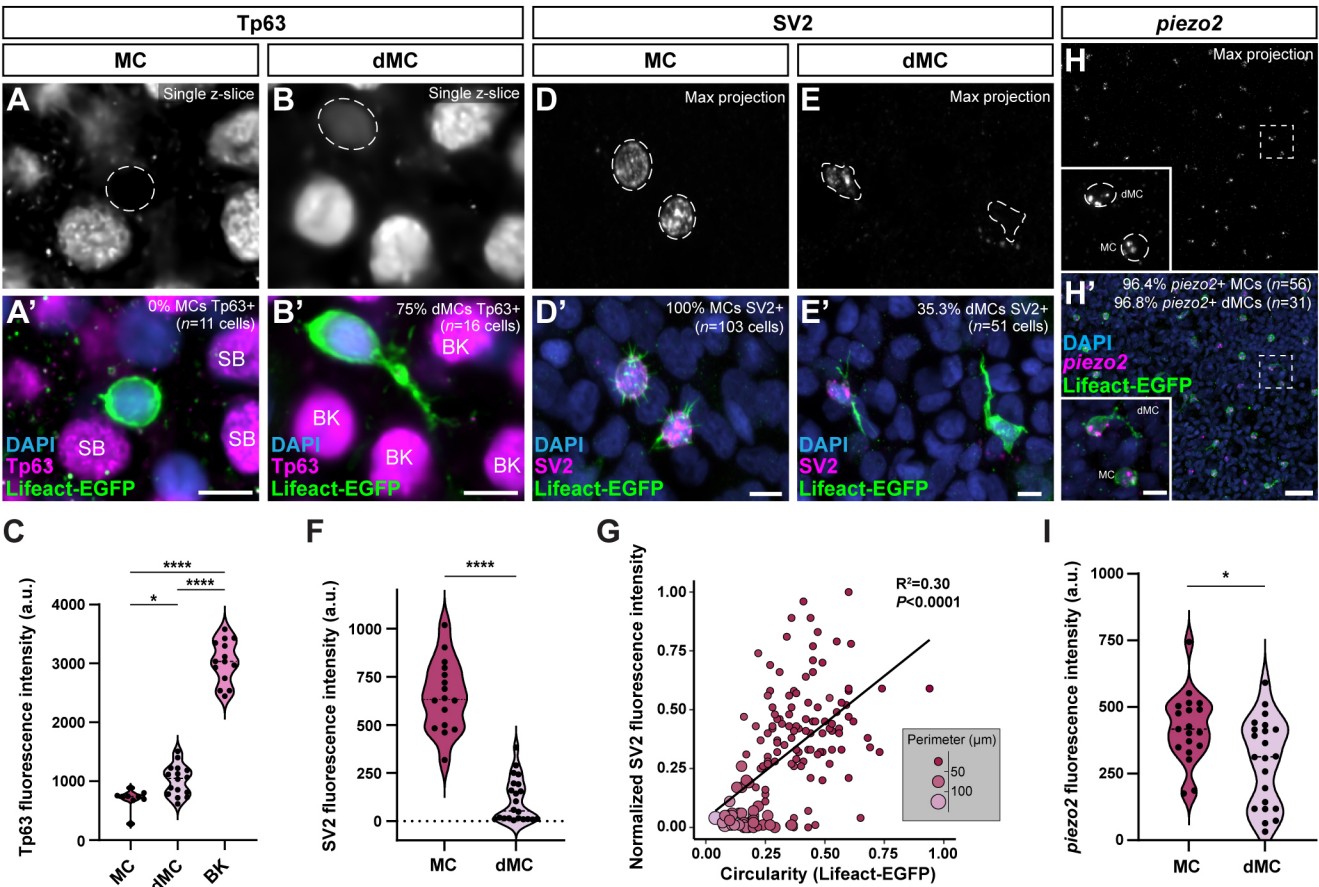

**Fig. 3. dMCs share molecular features with keratinocytes and MCs.** (A-B′) Representative images of MCs and dMCs within the 5 dpp regenerating scale epidermis of a *Tg(atoh1a:lifeact-egfp)* adult visualized with an anti-Tp63 (magenta) antibody. Dashed lines outline the nuclei. SB, suprabasal keratinocytes; BK, basal keratinocytes. (C) Violin plots of Tp63 staining intensity in MCs, dMCs and basal keratinocytes (BKs). Each dot represents a cell (*n*=9 MCs,16 dMCs and 14 basal keratinocytes from three fish). A one-way ANOVA with post-hoc Tukey HSD test was used to compare between cell types. (D-E′) Representative images showing MCs and dMCs within the 5 dpp regenerating scale epidermis of a *Tg(atoh1a:lifeact-egfp)* adult stained with anti-SV2 (magenta) and anti-GFP (green) antibodies. Dashed lines outline the cell bodies. (F) Violin plots of SV2 staining intensity in MCs and dMCs. Each dot represents a cell (*n*=16 MCs and 22 dMCs from ≥7 fish). A non-parametric Mann–Whitney test was used to compare between cell types. (G) Bubble plot of the correlation between the circularity of *Tg(atoh1a:lifeact-egfp)*-expressing cells and normalized anti-SV2 staining intensity at 5 dpp. Bubble size and color represent the perimeter in μm, as shown in the key. Each bubble represents a cell (*n*=153 cells from five fish). (H,H′) Representative images of MCs and dMCs within the 5 dpp regenerating scale epidermis of a *Tg(atoh1a:lifeact-egfp)* adult stained with an anti-*piezo2* HCR probeset (magenta) and an anti-GFP (green) antibody. Outlined areas are shown at higher magnification in the insets. (I) Violin plots of *piezo2* HCR staining intensity in MCs and dMCs. Each dot represents a cell (*n*=19 MCs and 22 dMCs from five fish). A non-parametric Mann–Whitney test was used to compare between cell types (**P*<0.05; *****P*<0.0001). Scale bars: 5 μm in A-B′,D-E′ and insets in H,H′; 20 μm in H,H′.

of Tp63 (Fig. 3A,A′). By contrast, dMCs exhibited significantly higher levels of Tp63 staining than MCs, albeit at lower levels than basal keratinocytes, with ∼75% of dMCs staining positive for Tp63 (Fig. 3B-C). Thus, dMCs express Tp63 at intermediate levels relative to basal keratinocytes and MCs.

Given our finding that dMCs expressed *atoh1a*, we postulated that dMCs might share additional molecular properties with MCs. MCs express markers of neuroendocrine and mechanosensory function, such as synaptic vesicle glycoprotein 2 (SV2) and the mechanosensitive cation channel Piezo2, respectively (Brown et al., 2023; Woo et al., 2014). Consistent with previous observations during ontogeny (Brown et al., 2023), we found that MCs associated with regenerating scales expressed SV2 (Fig. 3D,D′). SV2 staining of dMCs in regenerating skin revealed a mix of SV2+ and SV2− dMCs (Fig. 3E,E′). We quantified SV2 fluorescence intensity from dMCs with detectable signal and found dMCs displayed lower SV2 levels than MCs (Fig. 3F). Combined analysis of SV2 staining intensity and *atoh1a+* cell circularity revealed a positive correlation (Fig. 3G). Hybridization chain reaction (HCR) staining for *piezo2* labeled both MCs and dMCs in regenerating skin (Fig. 3H,H′), with lower levels of staining in dMCs compared to MCs (Fig. 3I). In summary, dMCs display molecular properties that overlap with both basal keratinocytes and MCs, supporting the interpretation that dMCs represent a transitional or immature MC rather than belonging to an alternative epidermal lineage.

### MCs and dMCs occupy different epidermal strata and have distinct polarities

Given the differences in morphologies between dMCs and MCs, we sought to compare the positions and polarities of these two populations. By analyzing reconstructed cross-sections in animals expressing a Cdh1 (E-cadherin) knock-in reporter to label keratinocyte membranes within the stratified epidermis (Cdh1-tdTomato; Cronan et al., 2018) (Fig. 4A), we found that MCs tended to reside below the upper keratinocyte layer (periderm), whereas dMCs were commonly in lower layers (Fig. 4B-D). To compare protrusion polarities, we fixed and stained scales with an anti-GFP antibody to visualize *Tg(atoh1a:lifeact-egfp)* and DAPI to label nuclei. The majority of MC microvilli extended in basal keratinocyte-facing or lateral directions, whereas periderm-facing processes were less common (Fig. 4E,G,H; Movie 1). By contrast, imaging of dMCs revealed long thin Lifeact-EGFP+ protrusions that routinely exceeded 10 μm in length and predominated in lateral orientations (Fig. 4F-H; Movie 2). Thus, our observations indicate MCs and dMCs occupy different strata of the epidermis, and extend actin-based protrusions that are of distinct sizes and polarities.

### dMCs display mesenchymal-like migration and dynamics

Apart from previous reports that tracked MC turnover over the course of days or weeks by imaging cytoplasmic or nuclear reporters (Brown et al., 2023; Wright et al., 2017), little is known about the *in vivo* dynamics of MCs. To compare the dynamics of the actin cytoskeleton in dMCs and MCs, we mounted and intubated *Tg(atoh1a:lifeact-egfp)* animals and performed live-cell confocal microscopy of fully intact skin over several hours. Imaging of MCs in adults revealed strong Lifeact-EGFP signal in microvilli, as expected (Fig. 5A, arrowhead). We captured numerous microvillar reorientation, retraction and extension events occurring in all MCs (Fig. 5A-A′′′′; Movie 3), suggesting previously unappreciated MC microvillar dynamics. MC microvilli rarely extended >3 μm from the cell body and often appeared to coalesce or merge together, although we lacked the resolution to reliably characterize these

events. Despite these microvillar dynamics, the MC cortex remained spherical during imaging (Fig. 5A-A′′′′). By contrast, dMCs dynamically rearranged both their protrusions and cell cortex (Fig. 5B-B′′′′; Movie 4). dMC protrusions tended to coalesce and extend from one end of the cell, with the opposite side having few or no protrusions (Fig. 5B′′).

Our observations of dMC protrusion dynamics and polarity led us to hypothesize that dMCs may migrate. Conversely, we hypothesized that MCs would be relatively immotile due to their association with somatosensory neurites and desmosomal contacts with keratinocytes (Hartschuh et al., 1986; Whitear, 1989). To characterize MC and dMC motility, we imaged fields of view from *Tg(atoh1a:lifeact-egfp)* juveniles containing MCs and dMCs over 5-6 h (Fig. 5C, Movie 5). Cell tracking revealed that MCs were largely immotile, with an average speed of 0.8 μm/h, which likely reflected imaging drift (Fig. 5C, insets; 5D). By contrast, dMCs migrated laterally within the epidermis with variable motility (Fig. 5C, insets; Fig. 5D). On average, dMCs migrated at 2.4 μm/h during imaging, but some moved at upwards of 7 μm/h, while others remained stationary (Fig. 5D). dMCs were most motile when adopting elongated ovoid cell bodies with long unipolar protrusions at one end of the cell, which correlated with the direction of migration in 95% of cells (*n*=20/21 cells from 9 fish) (Fig. 5E, pink arrowheads). By contrast, dMCs with multipolar protrusions were largely immotile (Fig. 5E, orange arrowheads). Although the majority of dMCs traveled in a persistent or linear manner (Fig. 5F), dMCs had a significantly reduced track displacement compared to total distance traveled (Fig. 5G), suggesting that dMCs migrate directionally but often switch directions. Together these observations indicate that MCs are immotile epithelial-like cells, whereas dMCs are motile mesenchymal-like cells.

MCs are generally considered post-mitotic (Merot and Saurat, 1988; Moll et al., 1996; Vaigot et al., 1987; Weber et al., 2023; Wright et al., 2017). Indeed, we did not observe instances of MC cell division in our live imaging of either homeostatic juvenile or regenerating adult skin (Table S1). By contrast, dMCs divided in both contexts (Fig. S2A-C; Table S1). During cell division, dMCs resorbed their protrusions, developed a smooth actin-rich cortex and underwent cytokinesis to generate two daughter dMCs, which established protrusions orthogonal to the division plane and migrated away from each other (Fig. S2A-C; Movie 6). Consistent with these observations, labeling of DNA synthesis during scale regeneration with 5-ethynyl-2′-deoxyuridine (EdU) failed to stain MCs, whereas 65% of dMCs stained positive (Fig. S2D-F). Thus, in addition to differences in motility, dMCs and MCs show contrasting cell cycle states.

### dMCs can directly mature into MCs

Given that dMCs exhibited characteristics of both keratinocytes and MCs (Fig. 3), we posited that dMCs may mature into MCs. Photoconversion of dMC nuclei labeled by *Tg(atoh1a:nls-eos)* would allow us to track individual cells longitudinally. Our finding that *de novo* generated *atoh1a+* cells expressing nls-Eos had ovoid nuclei at stages of scale regeneration when dMCs predominate (Fig. 2G,O′) suggested that nuclear morphology may distinguish dMCs from MCs. To assess this directly, we sought to create a nuclear reporter to use with *Tg(atoh1a:lifeact-egfp)* to compare nuclear and membrane morphologies. We previously found that zebrafish MCs express SRY-box transcription factor 2 (Sox2) (Brown et al., 2023), which functions at early stages of murine MC development (Perdigoto et al., 2014), making it an attractive target for reporter generation. Using a previous approach (Shin et al.,

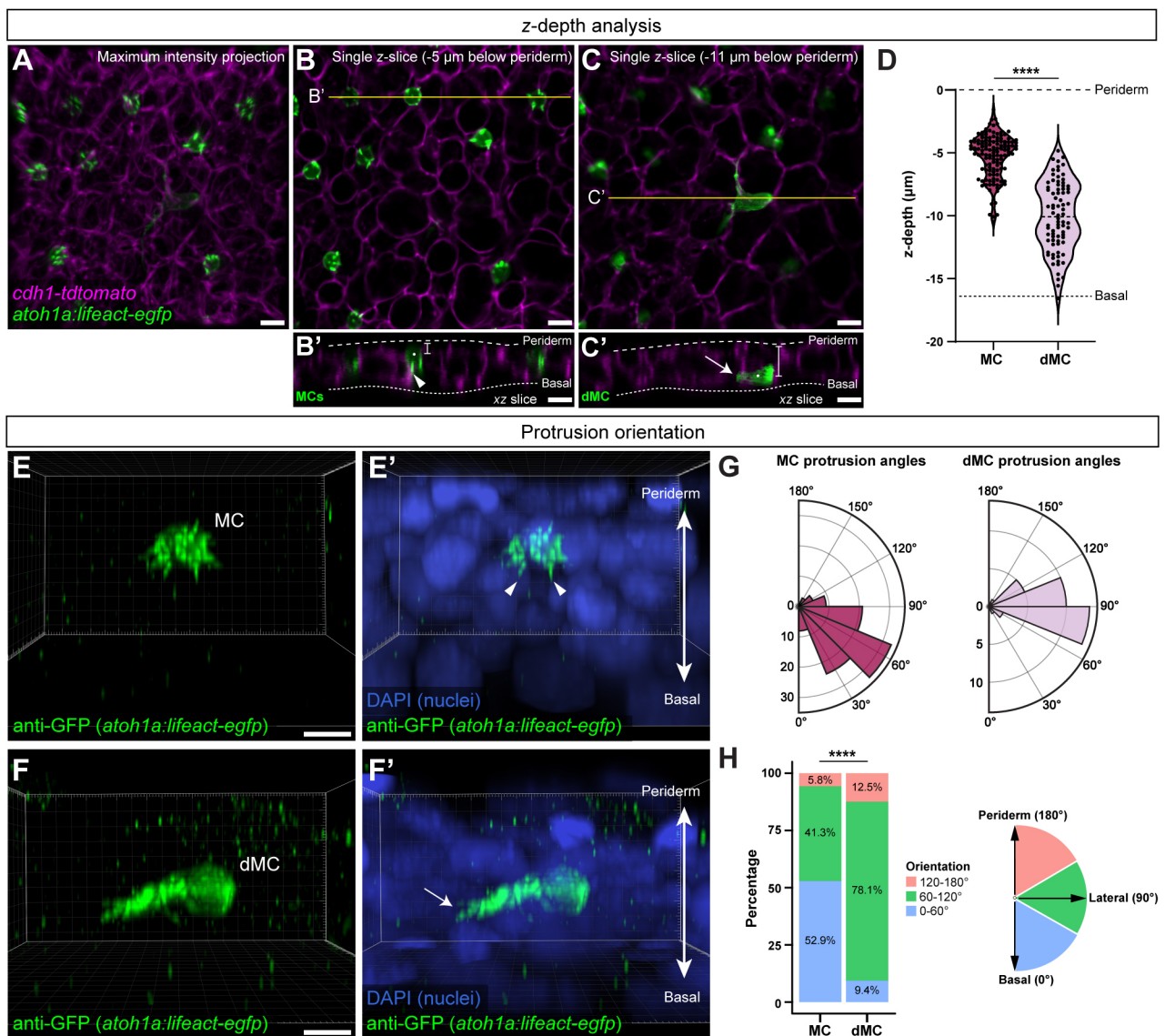

**Fig. 4. MCs and dMCs occupy different epidermal strata and have distinct actin polarities.** (A-C) Representative maximum intensity projection (A) or individual *z*-slices (B,C) of the scale epidermis. *Tg(atoh1a:lifeact-egfp)* labels dMCs and MCs (green), and *(cdh1-tdTomato)* labels keratinocyte membranes (magenta). (B′,C′) Reconstructed *xz* slices along the yellow lines in B and C. Dashed lines indicate the outer and inner epidermal margins. The MCs in B′ are located near the periderm layer of the skin with basal-facing protrusions (arrowhead), whereas the dMC in C′ is located near the basal layer of the skin with a lateral-facing protrusion (arrow). White dots and brackets indicate *z*-depth measurements used in D. Brightness and contrast have been adjusted in C and C′ to better illustrate dMC morphology. (D) Violin plots of *z*-depths of MCs and dMCs measured relative to the periderm surface of 130 MCs and 80 dMCs from five fish (11-12 mm SL). The lower dashed line indicates the average depth of the basal surface of basal keratinocytes in the data set (−16.4 µm). A non-parametric Mann–Whitney test was used to compare differences between cell types (****$P<0.0001$). (E-F′) Representative 3D reconstructions from *z*-stacks of a MC (E,E′) and a dMC (F,F′) stained with anti-GFP to label *Tg(atoh1a:lifeact-egfp)* and DAPI. Arrowheads indicate basal-facing microvilli; arrow indicates a laterally directed protrusion (see also Movies 1 and 2). (G) Polar histograms of MC and dMC protrusion angles. Lifeact-EGFP+ protrusions that could be individually resolved in 3D were measured relative to the *z*-axis of the epidermis, as diagrammed in H. Plots show 104 protrusions from 10 MCs (eight fish) and 32 protrusions from 11 dMCs (five fish). (H) Stacked bar charts depicting results in F, with protrusions binned based on orientation. A $\chi^2$ test was used to compare between cell types (****$P<0.0001$; $\chi^2$ statistic, 18.9903). Scale bars: 5 µm in A-C′,E,F.

2014), we created a *sox2-p2a-2x-sfCherry-nls* knock-in by inserting a cassette immediately upstream of the endogenous *sox2* stop codon containing a P2A peptide, followed by a tandem repeat of the rapidly maturing red fluorescent protein sfCherry (Nguyen et al., 2013) fused to a nuclear localization sequence (Fig. S3A,B). *sox2-p2a-2x-sfCherry-nls* larvae exhibited nuclear signal in known Sox2-expressing cell types, including within the inner ear and posterior lateral line neuromasts (Fig. S3C) (Hernández et al., 2007; Millimaki et al., 2010). In adult skin, 100% of *atoh1a+* dMCs and MCs expressed *sox2-2x-sfCherry-nls* (Fig. S3D,D′). As expected,

MCs had spherical nuclei, whereas dMCs had ovoid nuclei (Fig. S3D″). We found a strong positive correlation between nuclear and membrane circularities of *atoh1a+* cells (Fig. S3E), suggesting that nuclear morphology can distinguish dMCs and MCs.

To use photoconversion as a test of dMC maturation, we photoconverted individual *Tg(atoh1a:nls-eos)+* cells during scale regeneration (Fig. 6A,B). We specifically aimed to photoconvert nuclei with dim and/or ovoid nls-Eos signal, as these likely represented dMCs. We then compared the nuclear circularity of individual cells at 0 and 24 h post-conversion (hpc) and found a

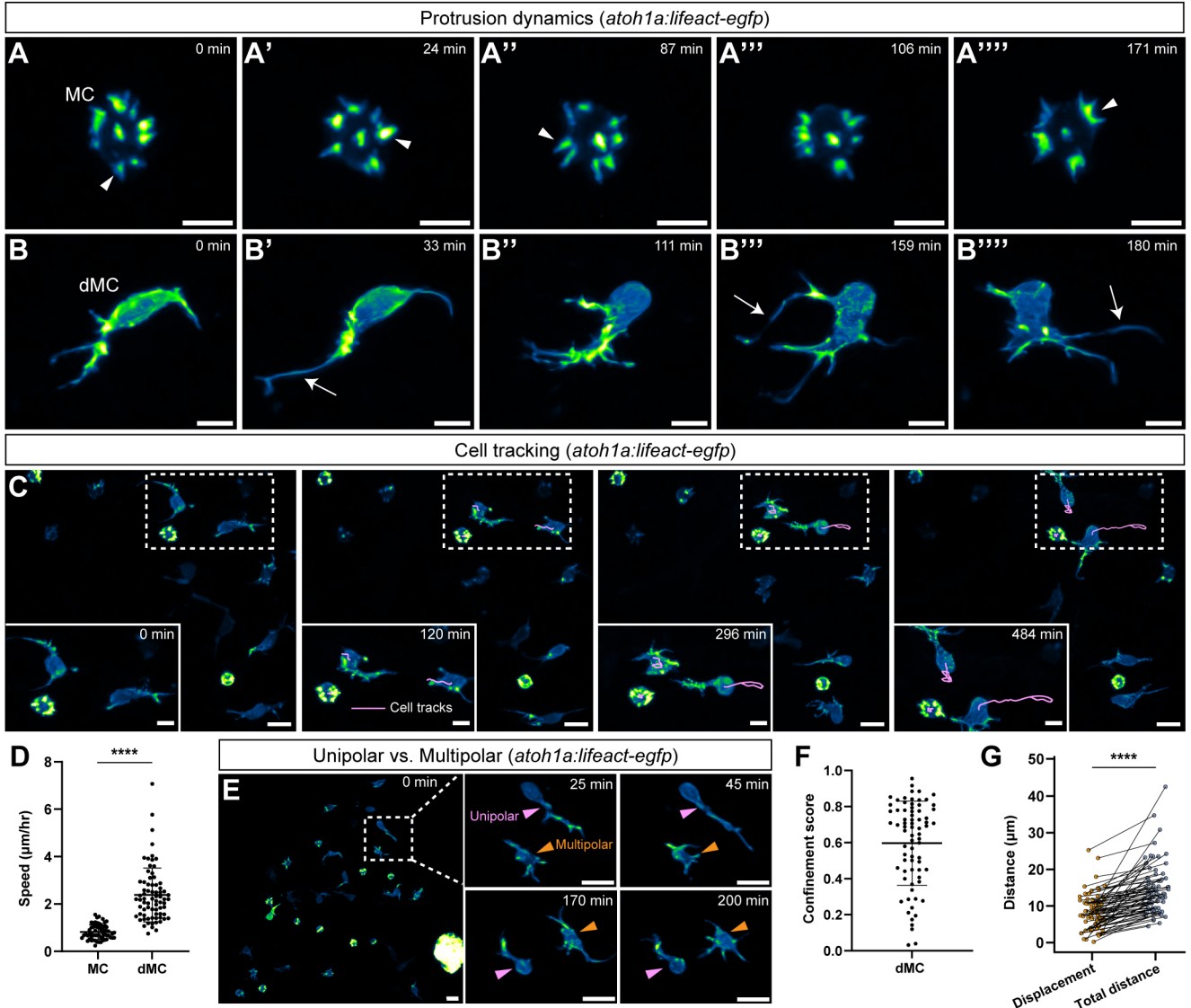

**Fig. 5. dMCs are motile cells with mesenchymal-like behaviors.** (A-A⁗) Time-lapse stills of a MC expressing *Tg(atoh1a:lifeact-egfp)*. White arrowheads indicate microvilli extension, retraction or merging events (see also Movie 3). (B-B⁗) Time-lapse stills of a dMC visualized with *Tg(atoh1a:lifeact-egfp)*. There are longer filopodial-like protrusions (white arrows) and an amorphous cell body (see also Movie 4). (C) Cell tracks (magenta) of individual MCs and dMCs over time (see also Movie 5). (D) Dot plot of cell speed of individual cells (*n*=60 MCs and 74 dMCs from four fish). A non-parametric Mann–Whitney test was used to compare between cell types (****P<0.0001). (E) Time-lapse stills from *Tg(atoh1a:lifeact-egfp)*-expressing juvenile skin. Magenta arrowheads indicate dMCs with unipolar protrusions. Orange arrowheads indicate dMCs with multipolar protrusions. Cells can switch between the unipolar and multipolar configurations. (F) Cell tracks scored for confinement ratio (*n*=74 dMCs from four fish). Values near 0 indicate confined movement and values near 1 indicate linear movement. (G) Paired dot plot of dMC track displacement, which measures the distance between the starting and ending point of each cell track, and total distance traveled (*n*=74 dMCs from four fish). A paired Mann–Whitney test was used to compare displacement and distance (****P<0.0001). In D and F, horizontal lines indicate the mean and error bars indicate the s.d. Scale bars: 5 µm in A-B⁗ and C, insets; 10 µm in C, E and E, insets.

significant increase in the circularity index at 24 hpc (Fig. 6C,D). By contrast, neighboring, non-photoconverted nuclei that began with a high circularity index (>0.9), which likely represented MCs, remained largely unchanged (Fig. 6D). These data are consistent with the hypothesis that dMCs can mature into MCs, although we note that this approach does not assess the detailed morphology of *atoh1a+* cells.

To directly track *atoh1a+* cell morphology over time, we examined our live-imaging dataset of juvenile and regenerating adult *Tg(atoh1a:lifeact-egfp)* skin. Consistent with the direct maturation hypothesis, we observed a small subset of dMCs (*n*=15 events in 8 fish) withdraw their long protrusions, round up their cell body and rapidly extend microvilli reminiscent of the

mature 'mace-like' MC morphology (Fig. 6E,F; Movies 7,8; Table S1). Based on these observations, we conclude that dMCs can directly mature into MCs during both development and regeneration of the trunk skin (Fig. 6G).

### Genetic loss of Eda results in altered dMC and MC morphologies in trunk skin

Eda signaling promotes the development of diverse vertebrate skin appendages (reviewed by Sadier et al., 2014), including zebrafish scales (Harris et al., 2008). During squamation, dermal cells express *eda*, whereas epidermal cells express *edar*, which encodes the Ectodysplasin A receptor (Harris et al., 2008; Aman et al., 2018). Zebrafish homozygous for a presumptive null *eda* allele

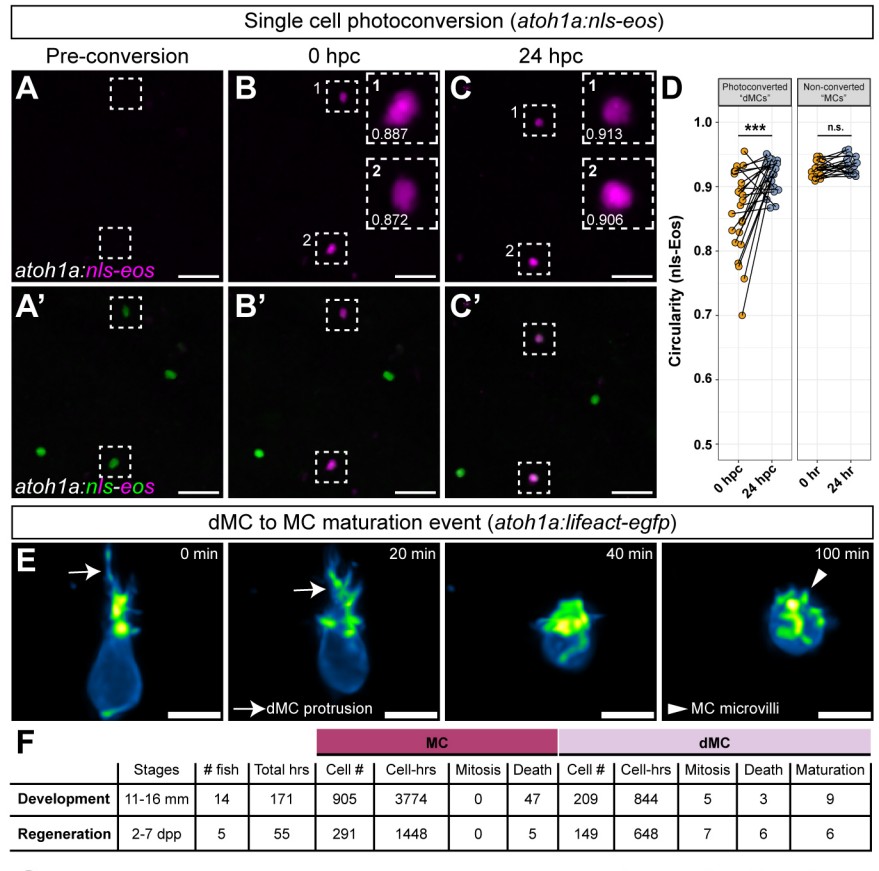

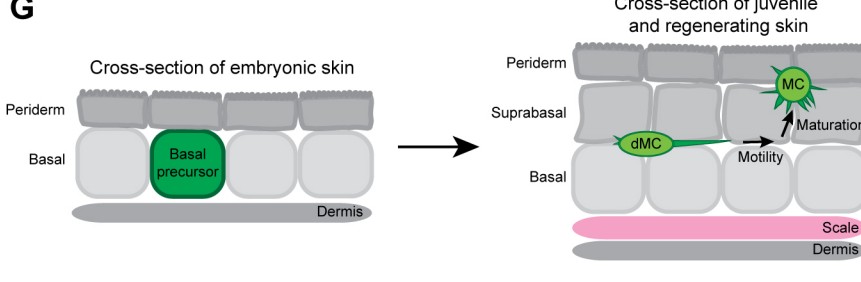

**Fig. 6. dMCs can directly mature into MCs in developing and regenerating zebrafish skin.** (A-C′) Representative images from a single *Tg(atoh1a:nls-eos)* fish before and after photoconversion of individual *atoh1a*+ nuclei during scale regeneration. Dashed outlines surround cells targeted for photoconversion. The insets in B and C contain the nuclear circularity index. (D) Left: paired dot plot of *atoh1a*+ nuclear circularity index at 0 and 24 h post-photoconversion (hpc) (*n*=24 cells total; four cells from two fish photoconverted at 4 dpp and re-imaged at 5 dpp; 20 cells from eight fish photoconverted at 5 dpp and re-imaged at 6 dpp). Right: paired dot plot of neighboring non-photoconverted MCs from the same cohort at 0 and 24 h (*n*=19 cells). A paired Mann–Whitney test was used to compare circularity values (***$P<0.001$; n.s., $P=0.06728$). (E) Time-lapse stills of a dMC to MC maturation event. Arrows indicate dMC protrusion retraction; arrowhead indicates formation of microvilli. The cell body transitions from an ovoid to a spherical shape. (F) Table summarizing observations from live imaging *Tg(atoh1a:lifeact-egfp)* during skin development and regeneration (see also Table S1 and Movie 7). (G) Schematic depicting the proposed model of dMC maturation events described in this study: (1) dMCs emerge in lower epidermal strata from *ΔNp63*+ embryonic basal keratinocyte progenitors; (2) dMCs migrate laterally in the direction of their protrusions; and (3) dMCs can directly adopt the mature MC morphology in upper epidermal strata. Scale bars: 20 μm in A-C′; 5 μm in E.

(*eda^dt1261/dt1261*, hereafter *eda^−/−*) display reduced MC density within trunk, but not facial, epidermis (Brown et al., 2023). To assess MC morphology in the absence of Eda signaling, we incrossed *eda^+/−* adults and used *Tg(atoh1a:lifeact-egfp)* localization to compare MC morphology between homozygous mutants and sibling controls. As expected, control trunk MCs developed a typical 'mace-like' morphology decorated by microvilli (Fig. 7A-A‴,G). By contrast, we observed a striking and penetrant loss of MC microvillar structures along the trunk of *eda^−/−* mutants (Fig. 7B-B‴,G). In *eda^−/−* trunk MCs, Lifeact-EGFP signal accumulated near the membrane in a ring-like fashion (Fig. 7B′, asterisk), reminiscent of dMC morphology during cell division (Fig. S2A-C). The few microvilli that formed in *eda^−/−* trunk MCs were short and thin, making them difficult to resolve with confocal microscopy. To determine whether Eda globally regulated MC morphology throughout the skin, we imaged MCs in the corneal epidermis, which is not squamated. This analysis revealed that corneal MCs elaborated microvilli in both *eda^−/−* mutants and siblings (Fig. 7C-D″,G). Finally, to determine whether loss of Eda signaling also impacted dMC morphology, we compared the maximum protrusion length of dMCs in *eda^−/−* mutants and controls, and found that mutant dMCs had significantly longer protrusions (Fig. 7E,

F,H; *eda^−/−* dMC mean length=16.3 μm, siblings=8.7 μm). In extreme cases, dMC protrusions reached almost 40 μm in length in *eda^−/−* trunk epidermis (Fig. 7F). Together, these results indicate that Eda is necessary for the normal morphologies of dMCs and MCs specifically within the trunk skin compartment.

We next questioned whether Eda regulated MC morphology during skin regeneration. As *eda* mutants lack scales, we sought an alternative injury model to scale pluck that would trigger MC regeneration. Mild injury can trigger MC regeneration in murine skin (Wright et al., 2017), and exfoliation of zebrafish skin initiates superficial keratinocyte regeneration (Chen et al., 2016). Thus, we began by assessing MC regeneration following exfoliation across a grid of adjacent scales with a cotton swab (Fig. 8A). In controls, we found that skin exfoliation resulted in a near complete loss of superficial keratinocytes, dMCs and MCs immediately following injury (Fig. 8B-D,K,L). At 4 and 7 days post-exfoliation (dpe), we observed restratification of the epidermis and a reappearance of dMCs and MCs (Fig. 8E,F,K,L). Most regenerated MCs adopted morphologies similar to those seen in uninjured skin, including the presence of microvilli (Fig. 8C,E,F,M). Similar to controls, exfoliation of the trunk of *eda^−/−* mutants resulted in a removal of epidermal layers, followed by restratification and a re-appearance of

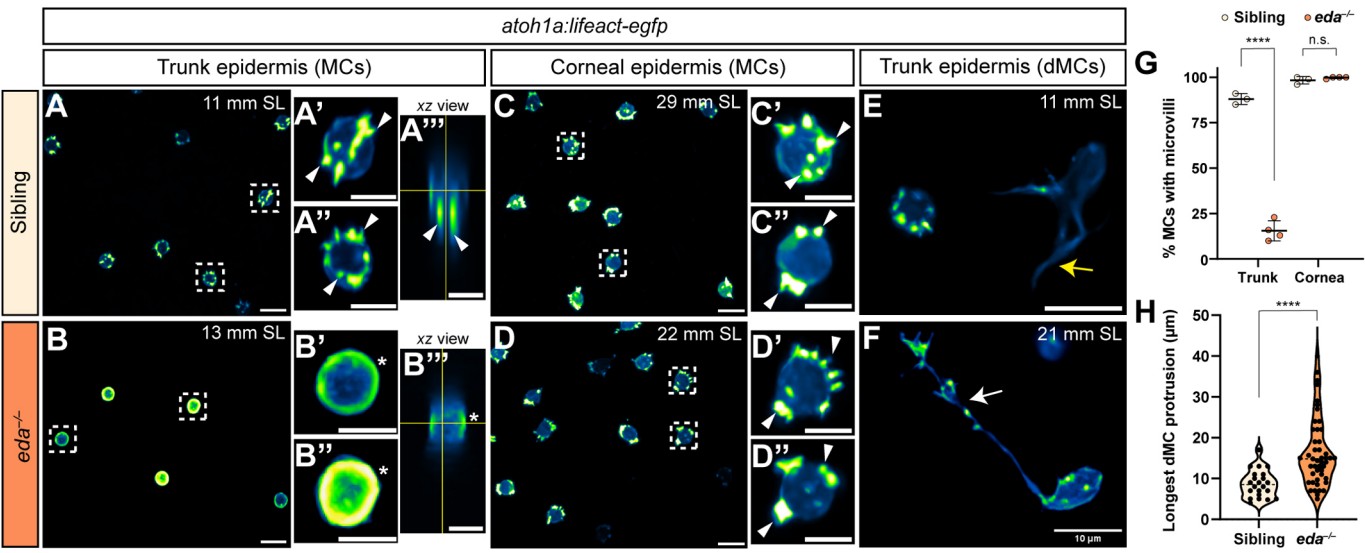

**Fig. 7. Genetic loss of Eda results in altered dMC and MC morphologies in trunk skin.** (A-B‴) Representative confocal images of MCs within the trunk epidermis of the indicated genotypes. Dashed boxes indicate cells magnified in A′,A″,B′,B″. Arrowheads indicate microvilli. Asterisks indicate intense Lifeact-EGFP signal forming a smooth ring-like cortical structure evident in cross-section in the $eda^{-/-}$ mutant epidermis. (A‴,B‴) $xz$ views of the cells shown in A′ and B′. (C-D″) Representative images of MCs within corneal epidermis of animals of the indicated genotypes. Dashed boxes indicate cells magnified in C′,C″,D′,D″. Arrowheads indicate microvilli. (E,F) Representative images of dMCs within the trunk epidermis of the indicated genotypes. Arrows indicate the longest protrusion on a dMC in eda sibling (E) or $eda^{-/-}$ mutant (F) epidermis. (G) Quantification of the percentage of MCs with discernable microvilli in trunk or corneal epidermis of the indicated genotypes. Each dot indicates an animal (9-20 mm SL) where a collection of images was analyzed (sibling trunk, n=386 cells from three fish; $eda^{-/-}$ trunk, n=235 cells from four fish; sibling cornea, n=169 cells from four fish; $eda^{-/-}$ cornea, n=332 cells from four fish). Fisher's exact test shows a significant difference between genotypes in the trunk but not cornea (****$P<0.0001$; n.s., $P=0.3413$). (H) Violin plots of the longest dMC Lifeact-EGFP+ protrusion within the trunk epidermis of juveniles of the indicated genotypes. Each dot represents a cell (siblings, n=22 dMCs from six fish; $eda^{-/-}$, n=47 dMCs from four fish). A non-parametric Mann–Whitney test (****$P<0.0001$) was used to compare between cell types. Scales bars: 10 μm in A-F; 5 μm in A′-A‴, B′-B‴,C′-C″,D′-D″.

dMCs and MCs at 4 and 7 dpe (Fig. 8G-L). However, a large proportion of regenerated MCs in $eda^{-/-}$ mutant skin contained a ring-like localization of Lifeact-EGFP and lacked microvilli (Fig. 8I,J,M), similar to our observations in uninjured skin. Thus, exfoliation provides a simple system for studying zebrafish MC regeneration independent of dermal appendage regeneration, and our results suggest that Eda promotes the formation of MC microvilli during epidermal regeneration.

## DISCUSSION

The defining morphological features of MCs allow their identification across diverse types of vertebrate skin (Hartschuh et al., 1986; Whitear, 1989). How MCs acquire this precise morphology during skin development or regeneration is unknown. Here, we directly address this knowledge gap by using the optical accessibility of zebrafish skin to track MC emergence during development and regeneration. Our results support a model in which dMCs emerge in lower epidermal strata and migrate locally within the epidermis. Their migration occurs both laterally and vertically within the epidermis, presumably due to their exploratory actin-rich membrane protrusions. As they mature in superficial strata, dMCs reorganize their actin cytoskeleton to elaborate polarized microvilli, thereby adopting the terminal MC morphology (Fig. 6G).

### A transient cell state during development and regeneration

We observed that dMCs appeared transiently during a period of skin organogenesis associated with dermal appendage growth and a rapid increase in MC density. These findings closely mirror changes in the 'dendritic:globular' MC ratio during human fetal palmar skin development (Kim and Holbrook, 1995). Furthermore, our lineage

tracing identified embryonic basal keratinocytes as dMC precursors, supporting previous ultrastructural findings that suggested basal keratinocytes give rise to 'transitional cells' with hybrid MC-keratinocyte properties (English, 1974, 1977; Garant et al., 1980; Tachibana and Ishizeki, 1981; Tachibana and Nawa, 1980). However, since those ultrastructural studies did not reconstruct transitional cell morphology, any relationship between transitional cells and dMCs observed by immunostaining (Kim and Holbrook, 1995; Moll et al., 1984; Nakafusa et al., 2006; Narisawa et al., 1993; Tachibana et al., 1997, 1998) has remained unclear. Based on our lineage tracing and morphological data, we propose that dMCs and transitional cells likely represent similar intermediates. We note, however, that dMCs may represent one of several intermediate states between basal keratinocytes and MCs, as previously suggested (Tachibana and Ishizeki, 1981).

Human MC regeneration after injury appears poor (Stella et al., 1994). Thus, studies of the mechanisms of MC regeneration in model organisms may inform improved wound-healing treatments. Previous work in mammalian models found that MCs regenerate after peripheral nerve crush (Burgess et al., 1974; Nurse et al., 1984), touch dome cauterization (Horch, 1982), full thickness wounding (Tachibana and Ishizeki, 1981) or skin shaving (Weber et al., 2023; Wright et al., 2017). We found that scale plucking induced a high proportion of dMCs during the early phase of scale regeneration, and that MC density ultimately recovered to pre-injury levels. Using photoconversion, we established that MCs in regenerated epidermis form predominantly from de novo regeneration rather than migration from surrounding epidermal reservoirs.

We used scale pluck to induce dMCs and measured their molecular properties during regeneration. dMCs expressed intermediate levels of the basal keratinocyte marker Tp63 and MC

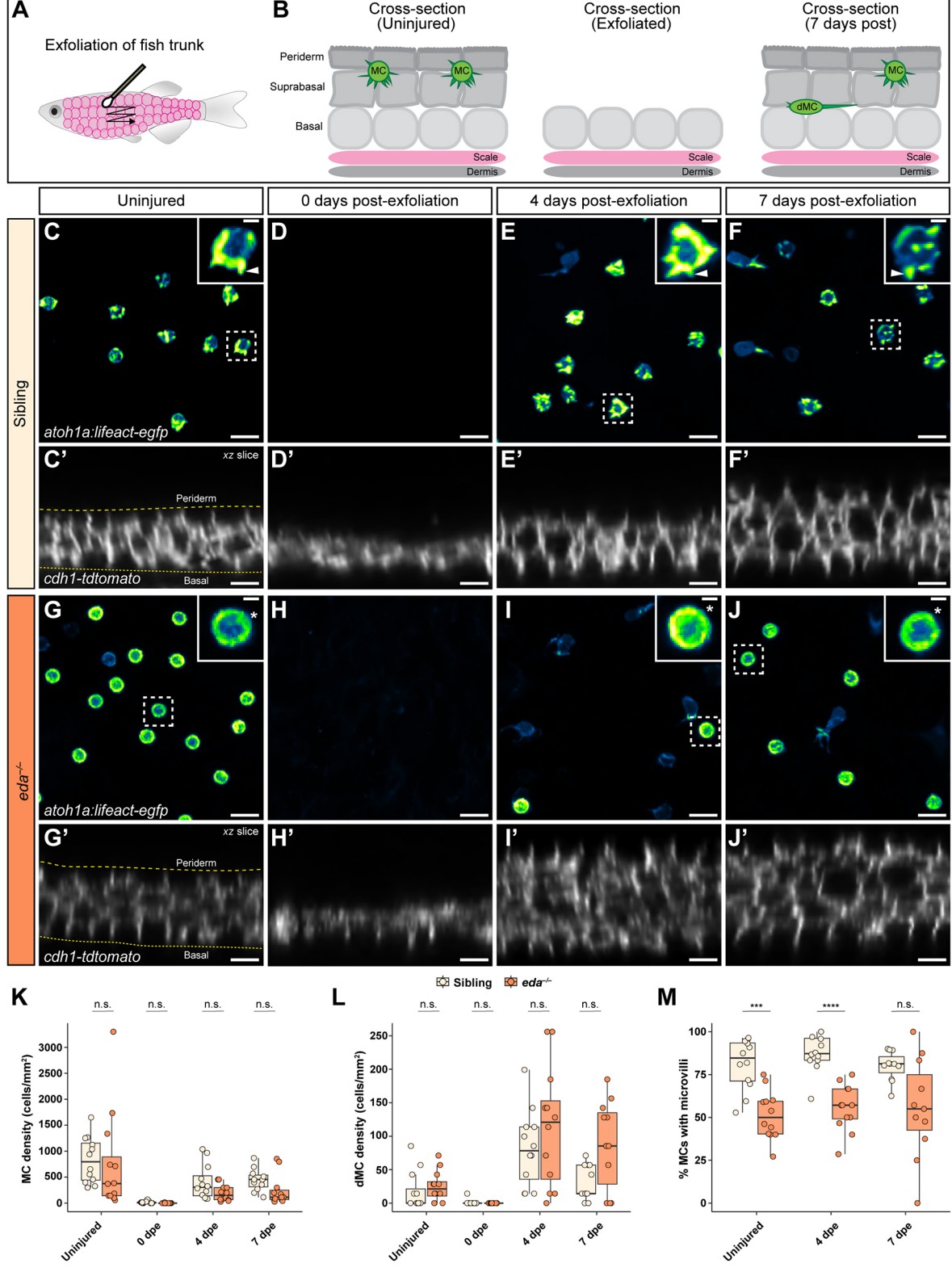

**Fig. 8.** See next page for legend.

markers *piezo2* and SV2. *piezo2* expression in dMCs suggests they may have the capacity for mechanosensory function. Consistent with this notion, we observed that most dMCs contacted axons and prior electrophysiological recordings found that mechanical stimulation resulted in rapid dMC depolarization (Maksimovic et al., 2014). Nevertheless, we cannot exclude additional possible functional roles for dMCs. Because our current analysis was restricted to a limited set of molecular markers, comparing the transcriptional profiles of keratinocytes, dMCs and MCs at a genome-wide scale is a worthy future endeavor.

**Fig. 8. Skin exfoliation induces MC regeneration and genetic loss of Eda results in altered MC morphology in regeneration.** (A) Schematic of the exfoliation method used to induce epidermal regeneration. (B) Schematic of the effects of exfoliation on the epidermis based on results in sibling fish. Exfoliation induces regeneration of superficial keratinocytes, dMCs and MCs by 7 days post-exfoliation (dpe). (C-J) Representative images of *Tg(atoh1a: lifeact-egfp)* within the adult trunk epidermis of the indicated genotypes. Dashed boxes indicate cells magnified in insets. Arrowheads indicate microvilli; asterisks indicate ring-like cortical Lifeact-EGFP signal. (C'-J') Reconstructed orthogonal slices of *(cdh1-tdTomato)* showing the epidermal structure of the indicated genotypes. Dashed lines in C' and G' indicate the outer and inner epidermal margins. (K,L) Box plots of the density of MCs (K) or dMCs (L) in sibling or $eda^{-/-}$ mutant skin during exfoliation-induced regeneration. (M) Box plots of the percentage of MCs with discernable microvilli in the trunk epidermis of animals of the indicated genotypes during regeneration. In K-M, each dot indicates an adult zebrafish (19-33 mm SL) where a collection of images were analyzed. Total *atoh1a+* cells analyzed: uninjured sibling trunk, $n=700$ cells from 12 fish; exfoliated (0 dpe) sibling trunk, $n=14$ cells from 12 fish; 4 dpe sibling trunk, $n=337$ cells from 12 fish; 7 dpe sibling trunk, $n=344$ cells from 11 fish; uninjured $eda^{-/-}$ trunk, $n=647$ cells from 12 fish; exfoliated (0 dpe) $eda^{-/-}$ trunk, $n=1$ cell from 12 fish; 4 dpe $eda^{-/-}$ trunk, $n=164$ cells from 12 fish; 7 dpe $eda^{-/-}$ trunk, $n=194$ cells from 11 fish. *P* values were determined by linear mixed-effects modeling with the Giesser-Greenhouse sphericity correction and Šidák's correction for multiple comparisons (****$P<0.0001$; ***$P<0.001$). Test results are reported in Table S2. In K-M, the boxes span the first and third quartiles, the horizontal line indicates the median and the whiskers extend to 1.5 times the interquartile range. Scale bars: 10 μm in C-J'; 2 μm in insets in C,E-G,I,J.

## Dynamic cellular behaviors of dMCs and MCs

Microvilli of several cell types exhibit dynamic motility (Cai et al., 2017; Meenderink et al., 2019). However, to our knowledge, no live-cell imaging of MC microvilli has been reported. Our *in vivo* imaging revealed that MC microvilli frequently extended, rearranged or retracted. Consistent with these dynamics, MC microvilli lack desmosomes, a feature noted across vertebrate skin compartments (Düring and Andres, 1976; Garant et al., 1980; Halata, 1975; Hashimoto, 1972; Iggo and Muir, 1969; Landmann and Halata, 1980; Takahashi-Iwanaga and Abe, 2001). We further found that zebrafish MC microvilli extend in lateral- or basal keratinocyte-facing orientations (Fig. 4B',E,G,H; Movie 1). Intriguingly, lamprey MC microvilli face both superficial and basal strata (Daghfous et al., 2020; Takahashi-Iwanaga and Abe, 2001), whereas MC microvilli within cat touch domes face superficial strata (Iggo and Muir, 1969). If microvilli promote MC function as previously proposed (Garant et al., 1980; Takahashi-Iwanaga, 2003; Takahashi-Iwanaga and Abe, 2001; Toyoshima et al., 1998; Yamashita et al., 1993), reconciling microvillar dynamics and polarity with MC function will be an interesting future avenue of research.

Our live imaging further revealed previously unrecognized dMC behaviors, many of which are properties of mesenchymal cells. First, we found that dMCs elaborate long dynamic actin-rich protrusions, which often extended from lateral-facing membranes, and were interdigitated between basal and suprabasal keratinocytes (Fig. 4C',F-H; Movie 2). Second, in contrast to immotile MCs, dMCs migrated laterally within the epidermis. dMC migration correlated with the presence of unipolar protrusions, whereas dMCs with multipolar protrusions were largely stationary. Intriguingly, Tachibana et al. (1998) described dMCs with similar shapes (termed unipolar and globular) in adult human oral mucosa. We speculate that the motile dMC intermediate serves to promote proper positioning of MCs – both laterally and vertically – within the epidermis to optimize sensory function. Third, we observed that dMCs could divide into two daughter cells with similar morphologies and behaviors. This suggests that at least a subset of dMCs may be transient amplifying

cells. Finally, and perhaps most significantly, our single-cell photoconversion and live-cell imaging indicate that dMCs serve as direct precursors to MCs. Although we do not know yet what cues this maturation, it appears to involve a transition from a mesenchymal-like to an epithelial-like state.

## Ectodysplasin signaling and MC maturation

We previously found that Eda signaling was partially necessary for MC development along the trunk (Brown et al., 2023). Here, we extended these studies by examining the morphology of dMCs and MCs in *eda* mutants. Surprisingly, whereas corneal MCs appeared unaffected, we found that most trunk MCs lacked microvilli in *eda* mutants, a previously unreported MC phenotype to our knowledge. This observation indicates Eda functions upstream of a MC microvillar program specifically within trunk skin, consistent with its known expression pattern (Aman et al., 2018; Harris et al., 2008). Dissecting the connections between Eda signaling, MC microvilli and MC function represent important areas for further study, and *eda* mutants offer a promising system to test MC sensory ability in the absence of microvilli. Our study does not address whether these Eda-dependent phenotypes result from direct effects of Eda signaling on the MC lineage or from indirect effects, e.g. through scale-derived signals.

In summary, we use the live imaging advantages of zebrafish to describe the dynamic cell behaviors of dMCs and MCs during development and regeneration. We establish that dMCs represent an intermediate cell state between basal keratinocytes and MCs. We further identify Eda signaling as essential for the regional morphogenesis of MCs. Finally, we speculate that the dMC intermediate state may have relevance for understanding the genesis of Merkel cell carcinoma. Our observations that dMCs can divide suggests they remain able to enter the cell cycle and, thus, could serve as the cell type of origin for Merkel cell carcinoma. Moreover, the mesenchymal-like behaviors of dMCs are reminiscent of metastatic cells. Consistent with these ideas, recent transcriptional analyses identified gene expression associated with a partial mesenchymal state in Merkel cell carcinoma tumor samples (Das et al., 2023; Karpinski et al., 2023).

## MATERIALS AND METHODS
### Animals
#### Zebrafish and developmental staging
Zebrafish were housed at 26-27°C on a 14/10 h light cycle. Published strains used were: AB (wild type), *(cdh1-tdTomato)*^xt18 (Cronan et al., 2018), *Tg(atoh1a:nls-eos)*^w214Tg (Pickett et al., 2018), *Tg(atoh1a:lifeact-egfp)*^w259Tg, *TgBAC(ΔNp63:Cre-ERT2)*^w267Tg (Brown et al., 2023), *Tg(actb2:LOXP-BFP-LOXP-DsRed)*^sd27Tg (Kobayashi et al., 2014), *Tg(Ola.Sp7:mCherry-Eco.NfsB)*^pd46Tg [referred to as *Tg(sp7:mCherry)*] (Singh et al., 2012), *Tg(Tru.P2rx3a:LEXA-VP16,4xLEXOP-mCherry)*^la207Tg [referred to as *Tg(p2rx3a:mCherry)*] (Palanca et al., 2013) and *eda*^dt1261 (Harris et al., 2008). Zebrafish of either sex were used. All zebrafish experiments were approved by the Institutional Animal Care and Use Committee at the University of Washington (protocol 4439-01).

To control for differences in growth rates, zebrafish post-embryonic development was staged based on SL (Parichy et al., 2009). The SLs of fish were measured using the IC Measure software (The Imaging Source) on images captured on a Stemi 508 stereoscope (Zeiss) equipped with a DFK 33UX264 camera (The Imaging Source). *eda* mutants and siblings were sorted by visible phenotype starting at 7 mm SL. Mutants were grown separately from siblings.

#### Generation of *(sox2-p2a-2x-sfCherry-nls)*^stl1034
The DNA sequence for *2x-sfCherry-nls* was synthesized (Integrated DNA Technologies) and inserted into a previously published *sox2-p2a* targeting

plasmid containing *sox2* homology arms (left arm, 1073 bp; right arm, 2036 bp) and a *p2a* sequence (Shin et al., 2014). The sequence 5′-TCACCTGTGAGATCCCCTAAGAAGAAGAGAAAGGTG-3′ was used to encode the nuclear localization signal. *sox2* TALEN RNAs have been previously published (Shin et al., 2014) and were synthesized with the SP6 mMessage mMachine Kit (ThermoFisher, AM1340). One-cell embryos were injected with *sox2* TALEN RNAs at 35 ng/µl each and *sox2-p2a-2x-sfCherry-nls* targeting plasmid, which was linearized by a NcoI digest, at 10 ng/µl in 1× injection buffer (0.1 M KCl and 0.003% Phenol Red). Embryos were screened for fluorescence under a stereomicroscope and those displaying the *sox2* expression pattern were raised to adulthood. G0 adults were crossed to AB to identify founders and obtain F1 progeny. A single knock-in line was established and designated *(sox2-p2a-2x-sfCherry-nls)^stl1034*. The insertion was validated by performing PCR with either PrimeSTAR GXL (Takara Bio, R050A) or Standard TAQ Polymerase (NEB, M0273L) on fin clip lysates using the primers: sR3, 5′-AGTGCTCCCTGACCCTTTGAGAGTCCG-3′; sF2, 5′-GTAACCC-CGCCCCTTTATGCAAACCG-3′; cF1, 5′-TGAGGCACGTCATTCTA-CAGGCG-3′; sFC, 5′-GGGCACAACAGGACCTAAGA-3′; and cR1, 5′-ATAGTCAGGGATGTCAGCGGGGT-3′, as shown in Fig. S3A,B.

### Induction of *TgBAC(ΔNp63:Cre-ERT2)* with 4-OHT
To activate recombination with Cre-ERT2, 1 dpf *TgBAC(ΔNp63:Cre-ERT2)*; *Tg(actb2:LOXP-BFP-LOXP-DsRed); Tg(atoh1a:lifeact-egfp)* embryos were treated with 10 µM 4-OHT for 24 h and screened for successful recombination at 3-5 dpf, as evidenced by DsRed+ epidermal cells. 4-OHT (MilliporeSigma, H7904) was prepared as described previously (Felker et al., 2016).

### Scale pluck
A scale pluck protocol was used either to collect scales for immunofluorescence or to induce scale regeneration. To prepare fish for scale pluck, fish were anesthetized in 0.006-0.012% buffered MS-222 (MilliporeSigma, E10521) diluted in system water until a surgical plane of anesthesia was reached. Anesthetized fish were moved under a dissecting microscope and placed on the lid of a petri dish. Dumont #5 forceps were used to pluck scales from the trunk in a posterior to anterior procession. Fish were then returned to system water and monitored for full recovery.

### Exfoliation
Prior to exfoliation, adults were anesthetized in 0.006-0.012% buffered MS-222 diluted in system water and held in place using a flexible restraint within a custom 3D-printed chamber. Epidermal exfoliation was performed by using gentle pressure to pass the tip of a cotton tipped applicator (Avantor, 76048-960) back and forth in a zigzag pattern across a region of the lateral trunk skin. This region spanned a grid of multiple scales with a maximum size of 70 mm². Care was taken not to dislodge scales during the treatment. Fish were recovered in system water following the procedure.

### Imaging
#### Confocal imaging
Short-term live imaging of juvenile and adult zebrafish was achieved by anesthetizing fish in 0.006-0.012% buffered MS-222 diluted in system water for ~5 min. Once the fish was immobilized, it was mounted in a custom imaging chamber and the fish body was secured by carefully adding molten 1% agarose in system water. Agarose was not applied to the trunk region where imaging occurred or near the gills and mouth of the fish to ensure survival. Agarose embedded fish were then covered with MS-222 solution and imaged using an A1R MP+ scanhead mounted on a Ni-E upright microscope (Nikon). Except where noted otherwise, a 16× water dipping objective (N.A. 0.8) was used for *z*-stack acquisition and images were post-processed using the Denoise.ai function in NIS-Elements (Nikon). For short-term imaging, fish were taken off the microscope after a maximum 25 min of image collection and returned to system water for recovery.

Long-term live imaging of juvenile zebrafish was achieved through an intubation-based protocol that delivered tricaine water to immobilized zebrafish using a peristaltic-pump, similar to Xu et al. (2015). In brief, fish

were anesthetized in buffered MS-222 for 8-10 min until gill movement became very slow. Fish were transferred to a custom imaging chamber and embedded with 1% agarose, as described above. Continuous delivery of buffered MS-222 was achieved by using forceps to gently insert polyethylene tubing (Becton Dickinson 427421 or 427400) into the mouth of the fish. This delivery line was held in place by modeling clay and 0.08% MS-222 was delivered to the fish by a peristaltic pump at a flow rate of 2-3 ml per min. Multipoint time lapse imaging was achieved through NIS-Elements by manually setting large *z*-stacks for each field of view to account for sample drift during imaging. Depending on the experiment, time lapse images were acquired every 3-5 min for up to 6 h at room temperature. To revive fish after multi-hour time lapse imaging, the peristaltic pump line was transferred to a bottle containing system water and fish were closely monitored for recovery. Data acquired from zebrafish that did not survive the intubation session were excluded from further analysis.

### Scale regeneration time course
Scales were plucked from adult *Tg(atoh1a:lifeact-egfp);Tg(sp7:mCherry)* zebrafish, as described above. To facilitate subsequent imaging, the two rows along the midline were plucked. Plucked fish were returned to the recirculating system and imaged using confocal microscopy at the indicated dpp. 1.5× and 8× images were collected for quantification and downstream analyses.

### Whole animal photoconversion
Prior to scale removal, *Tg(atoh1a:nls-eos)* zebrafish were exposed to light from a UV LED flashlight (McDoer) for 15 min in a reflective chamber constructed from a styrofoam box lined with aluminum foil. A similar lateral region of the trunk was imaged over subsequent days identified by approximate body position below the dorsal fin and relative to underlying pigment stripes.

### Single-cell photoconversion
Prior to photoconversion of individual cells from *Tg(atoh1a:nls-eos)* zebrafish, scales were removed to induce regeneration. At 4 (*n*=2 fish) or 5 (*n*=8 fish) dpp, fish were anesthetized in MS-222, mounted and intubated as described above. Ovular and/or dim *atoh1a*+ nuclei were targeted for photoconversion using the stimulation program of NIS-Elements with a 405 nm laser at 5% power for 3 s. Stimulation regions were squares with edge lengths of 1.5-2 µm and centered on *atoh1a*+ nuclei. Fish were imaged immediately after photoconversion, as well as at 24 h post conversion.

### Staining
#### Alizarin Red S staining
Alizarin Red S stains calcium deposits, allowing visualization of osteoblast-derived zebrafish scales. To visualize zebrafish scale development, live animals were stained for 20 min in a solution of 0.01% (wt/vol) Alizarin Red S (ACROS Organics, 400480250) dissolved in system water and shielded from light, rinsed three times for 5 min each in system water, then transferred back into fresh system water as described previously (Bensimon-Brito et al., 2016).

#### Immunofluorescence
Zebrafish were anesthetized in a solution of 0.012% MS-222 in system water for 2 min. Approximately 30 scales were plucked and transferred to 1.5 ml tubes containing 375 µl 1×PBS. For fixation, 125 µl of 16% paraformaldehyde (PFA) was added to achieve a final 4% PFA solution. Scales were incubated for 20 min at room temperature on a gently rotating platform. PFA was washed out by carefully removing the fixation solution and replacing it with 400 µl of 0.2% PBST (1×PBS+0.2% Triton X-100). Scales were washed three times for 5 min each with PBST then blocked with blocking solution (10% normal goat serum in PBST) for 2-3 h at room temperature. Blocking solution was removed and 200 µl of primary antibody solution made up in blocking solution was added. Primary antibodies used were: mouse monoclonal anti-SV2 (DSHB, SV2, RRID: AB_2315387) at 1:50, rabbit polyclonal anti-Tp63 (GeneTex, GTX124660, RRID:AB_11175363) at 1:800 or rabbit polyclonal anti-GFP (Thermo Fisher Scientific, A11122, RRID:AB_221569) at 1:500. Scales were

incubated in primary antibody overnight at 4°C, protected from light. The next day, scales were washed four times for 15 min each with PBST before adding secondary antibody made up in blocking solution. Secondary antibodies used were: goat anti-mouse Alexa Fluor 647 (Thermo Fisher Scientific, A32728, RRID:AB_2633277) at 1:500 or goat anti-rabbit Alexa Fluor 488 (Thermo Fisher Scientific, A32731, RRID:AB_2633280) at 1:1000. Scales were incubated for 2 h at room temperature, protected from light. Secondary solution was washed out four times for 15 min each using PBST. To visualize nuclei, 5 ng/µl DAPI (MilliporeSigma, 508741) was added. DAPI was washed out four times for 5 min each using PBST. Scales were mounted epidermis-side up between a microscope slide and coverslip in ProLong Gold (Thermo Fisher Scientific, P36930). Imaging was performed with a 40× (NA 1.3) oil immersion objective.

### Hybridization chain reaction
HCR on adult zebrafish scales using a custom *piezo2* (accession number, XM_021468270.1; set size: 20; amplifier: B3) probe set was performed as described previously (Brown et al., 2023). Scales were imaged with a 25× water immersion objective (N.A. 1.1).

### EdU labeling during scale regeneration
Scales were plucked from adult *Tg(atoh1a:lifeact-egfp);Tg(sp7:mCherry)* zebrafish to induce scale regeneration. Fish were returned to their tanks for recovery prior to EdU injection. Regenerating fish were anesthetized in MS-222 and placed ventral-side up on a sponge situated under a dissecting microscope. 10 µl of 10 mM EdU (Thermo Fisher Scientific, C10640) was injected intraperitoneally in the region between the pelvic fins. Fish were transferred back to system water for recovery. Approximately 20 h later, these fish had their regenerating scales plucked and subjected to EdU staining following the manufacturer's instructions. If downstream immunofluorescence was required, this was performed after EdU detection. Stained scales were imaged with a 40× (NA 1.3) oil immersion objective.

### Image and statistical analysis
#### Image processing
Image processing was performed using FIJI/ImageJ (Schindelin et al., 2012) or Imaris (Oxford Instruments). Unless otherwise indicated, all images were gathered through z-stack acquisition and displayed as maximum intensity projections.

#### Cell counting and shape analysis
*atoh1a+* cells were classified as dMCs or MCs as follows: Confocal z-stacks were max projected and the ImageJ 'Cell Counter' function was used to manually classify *atoh1a+* cells as dMCs (oblong cell body and one or more filopodial-like protrusions) or MCs (spherical cell body and microvilli-like protrusions) based on cell morphology. For the shape analysis in Fig. 1L, max projected images were thresholded using Huang's algorithm. Using the 'Analyze Particles…' function, circularity $\left(4\pi \times \frac{[Area]}{[Perimeter]^2}\right)$ and roundness $\left(4 \times \frac{[Area]}{\pi \times [Major\ axis]^2}\right)$ were calculated. In Fig. 1N, 'circular' cells were defined as those having circularity and roundness values >0.7. For the shape analysis in Fig. 3G, confocal z-stacks were 3D projected and subjected to thresholding in ImageJ using Huang's algorithm to outline each *Tg(atoh1a: lifeact-egfp)+* cell. SV2 fluorescence intensity was measured as the mean gray value within the outlined cells using the 'Analyze Particles' feature of ImageJ. SV2 fluorescence intensity values were normalized first by subtracting mean background fluorescence and then to the minimum and maximum mean gray value among all the cells. For measuring cell shape, the outlined cells were analyzed in ImageJ as in Fig. 1L for circularity. The 'Analyze Particles' feature was used to measure the perimeter of each outlined cell.

#### Fluorescence intensity analysis
Fluorescence intensity of MCs and dMCs was calculated from confocal z-stacks acquired using identical settings by corrected total cell fluorescence (CTCF) in ImageJ as follows:

$$CTCF = Integrated\ Density$$
$$- (Area\ Of\ Selected\ Cell \times Mean\ Background\ Fluorescence).$$

#### Z-depth analysis
For the z-depth analysis in Fig. 4D, confocal z-stacks were acquired using a 25× water dipping objective (N.A. 1.1) and a z-step of 0.3 µm. The outer surface of the periderm was defined as a depth of 0. The distance from the center of the cell bodies of dMCs and MCs to the periderm surface was measured on reconstructed yz slices using the line tool in ImageJ.

#### Cell tracking
To quantify cell motility metrics, the TrackMate plugin in ImageJ was used (Tinevez et al., 2017). Z-stack acquired time-lapse images were imported and made into maximum intensity projections. To correct xy drift encountered during long-term imaging acquisition, the ImageJ plugin 'correct 3D drift' was used. Manual detection in TrackMate was performed by tracking individual cells of interest. The metrics calculated from TrackMate (speed, total distance, displacement and confinement score) are detailed at https://imagej.net/plugins/trackmate/analyzers/.

#### Recombination efficiency
To calculate the recombination rate, confocal z-stacks were acquired of the trunk epidermis of *TgBAC(ΔNp63:Cre-ERT2); Tg(actb2:LOXP-BFP-LOXP-DsRed); Tg(atoh1a:lifeact-egfp)* animals at 1-2 months post-fertilization that had been treated with 4-OHT at 1 dpf. Maximum intensity projections were generated and the BFP and DsRed channels thresholded. The '%Area' of the thresholded channels was determined using the 'Measure' function in ImageJ. The overall recombination efficiency was calculated as:

$$\frac{\%Area_{DsRed}}{\%Area_{BFP} + \%Area_{DsRed}}.$$

#### Innervation quantification
To measure innervation frequency in Fig. S1, confocal z-stacks acquired from *Tg(atoh1a:lifeact-egfp); Tg(p2rx3a:mCherry)* trunk skin were analyzed using a previously described ImageJ macro (Brown et al., 2023).

#### Statistical analysis
Statistical analysis and graphing were performed using R (R Core Team, 2023), MATLAB (MathWorks) or GraphPad (Prism). Statistical tests used and sample numbers are described in the corresponding figure legends. The $\chi^2$ and Fisher's exact tests were performed on raw count matrices. The horizontal dashed line on violin plots indicates the median.

### Acknowledgements
We thank the LSB Aquatics staff for animal care, Dr David Tobin for sharing zebrafish and Matt Footer for creating the exfoliation chamber. The authors are grateful to all members of the Rasmussen lab for discussion, technical assistance and support. The SV2 monoclonal antibody used in this study was obtained from the Developmental Studies Hybridoma Bank, created by the NICHD of the NIH and maintained at The University of Iowa, Department of Biology, Iowa City, IA 52242.

### Competing interests
The authors declare no competing or financial interests.

### Author contributions
Conceptualization: E.W.C., J.P.R.; Formal analysis: E.J.A.Q.; Funding acquisition: E.W.C., E.C.B., S.Z.F., S.M.S., L.S.-K., J.P.R.; Investigation: E.W.C., E.C.B., S.Z.F., A.S.F., C.E.A.G., S.M.S., E.J.A.Q., A.A.S., N.G.Y., J.P.R.; Methodology: E.W.C., E.C.B., S.Z.F., A.S.F., C.E.A.G., S.M.S., E.J.A.Q., A.A.S., N.G.Y., J.S., J.P.R.; Resources: J.S.; Supervision: L.S.-K., J.P.R.; Validation: J.P.R.; Visualization: E.W.C., E.C.B., S.Z.F., A.S.F., S.M.S., J.P.R.; Writing – original draft: E.W.C., J.P.R.; Writing – review & editing: E.C.B., S.Z.F., A.S.F., S.M.S., J.P.R.

## Funding

This work was funded, in part, by Graduate Research Fellowships from the National Science Foundation to E.W.C. and S.Z.F., by an Institute for Stem Cell and Regenerative Medicine Graduate Fellowship to E.C.B., by a Cell and Molecular Biology Training Grant (T32 GM136534 to S.M.S.), by the National Institute of General Medical Sciences (R35 GM118179 to L.S.-K.), by the Eunice Kennedy Shriver National Institute of Child Health and Human Development (R01 HD107108 to J.P.R.), and by a New Investigator Award to J.P.R. from the Fred Hutch /University of Washington/Seattle Children's Cancer Consortium, which is supported by the National Institutes of Health/National Cancer Institute Cancer Center Support Grant P30 CA015704. J.P.R. is a Washington Research Foundation Distinguished Investigator. Open Access funding provided by the University of Washington. Deposited in PMC for immediate release.

## Data and resource availability

All relevant data and details of resources can be found within the article and its supplementary information.

## Peer review history

The peer review history is available online at https://journals.biologists.com/dev/lookup/doi/10.1242/dev.204810.reviewer-comments.pdf

## Special Issue

This article is part of the Special Issue 'Lifelong Development: the Maintenance, Regeneration and Plasticity of Tissues', edited by Meritxell Huch and Mansi Srivastava. See related articles at https://journals.biologists.com/dev/issue/152/20.

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
