## [Peer Review File · Development (Cambridge, England)]

Dendritic *atoh1a*⁺ cells serve as Merkel cell precursors during skin development and regeneration

Evan W. Craig, Erik C. Black, Samantha Z. Fernandes, Ahlan S. Ferdous, Camille E. A. Goo, Sheridan M. Sargent, Elgene J. A. Quitevis, Avery Angell Swearer, Nathaniel G. Yee, Jimann Shin, Lilianna Solnica-Krezel and Jeffrey P. Rasmussen
DOI: 10.1242/dev.204810

Editor: Kenneth Poss

Review timeline

Submission to Review Commons: 25 September 2023
Submission to Development: 25 March 2025
Accepted: 4 May 2025

Reviewer 1:

Evidence, reproducibility and clarity

****Summary:****

In this manuscript, the authors use confocal imaging techniques to morphologically characterize Merkel cells during their maturation process in the zebrafish skin. Using an Factin reporter, they identify two morphologically distinct populations of *atoh1a*⁺ cells: 1) Mature Merkel cells (MCs), which had previously been described in zebrafish, and 2) a transient population sharing morphological characteristics with so called dendritic Merkel Cells (dMCs), that were described in mice and humans but not previously identified in zebrafish. It was unknown whether dMCs represent a developmentally immature MC state or a functionally distinct subpopulation of neuroendocrine cells. The authors go on to show that dMCs represent the primary *atoh1a*⁺ cell type during skin regeneration and share features of both basal keratinocytes and Merkel cells, leading them to speculate that they could be MC precursors. Confocal time lapse imaging further showed that MCs and dMCs differ in the polarity of their protrusions. In some of the lapses, dMC can be seen maturing into MCs, providing evidence that they could be precursor cells. MC to dMC reversion events are also observed, albeit less often. Finally, the authors show that loss Ectodysplasin A (*Eda*) signaling disrupts MC microvilli formation, identifying this pathway as a potential regulator of MC morphology.

****Major comments:****

- The authors conclude that dMCs represent an intermediate state in the MC maturation program. This is based on the observation that the percentage of dMCs decreases over time and the fact that they share characteristics of both keratinocytes and MCs. In addition, dMCs are observed to mature into MCs in time lapses. However, these findings do not completely rule out the possibility that dMCs represent a transient, functionally distinct population of MCs. The authors should discuss this possibility. Additionally, some clarifications on the data could help strengthen their conclusion:
- Figure 1 I-K: The interpretation of the simultaneous increase of dMCs and MCs is not clear. Shouldn't the percent of dMCs be highest at 8-9mm and then go down, when MCs first start to appear?
- Fig. 2K: These results could also mean that dMCs numbers stay the same and only MCs increase in number. Does not imply lineage as stated in line 182 where the authors say that dMCs are a transient population. Please also report the total number of dMCs.
- Figure 5 F and G: In these time lapses, "a small subset of dMCs (n>10)" is observed to adopt MC morphology. Does this mean 10 cells, and if so, out of how many? The authors should clarify how many time lapses were taken, and quantify the percentage of dMCs undergoing this process. The

same goes for the reciprocal process, MC to dMC conversion, which happens only "in rare instances (n=2)".

- Use photoconversion of single cells to establish lineage relationship. The 2 time lapses shown are not statistically significant and the identity of MCs in these movies is solely based on morphology.
- In the last part of the paper, the authors show that trunk dMCs and MCs adopt abnormal morphologies in the absence of Eda signaling. However, this phenotype is not seen in the corneal epidermis, which is not squamated. Since Eda mutants do not develop scales, could the altered morphology in the trunk be due to the absence of scales? If possible, the authors should inhibit Eda signaling after the formation of scales or tone down their conclusions.
- Line 264: The authors write: 'Consistent with this notion, dMC-dMC or dMC-MC contacts resulted in lateral dMC movement away from the contact (Movie 4). Together these observations suggest that MCs are immotile, epithelial-like cells, whereas dMCs are motile, mesenchymal-like cells that undergo contact inhibition upon encountering another atoh1a+cell'. The lateral movement of dMCs after contacting MCs needs to be quantified before it can be interpreted as contact inhibition.

****Minor comments:****

- 'Defects in the morphogenesis of actin-based protrusions are linked to a variety of diseases, including colorectal cancer and deafness'. Please provide refs.
- Line 145: this experiment does not show motility. Just that basal keratinocytes give rise to them.
- Line 165. Cells increase by 14dpp and do not seem to plateau at 7dpp. Please discuss.
- Line 190. Does Figure 3A not show basal keratinocytes? Only Figure 3B is cited.
- Figure 3: Within individual cells, is there a negative correlation between SV2 staining and tp63 staining in dMCs? Or between sphericity and tp63 staining?
- If dMCs are immature, are they already innervated by somatosensory axons?
- Line 284: Indeed, during our live-imaging of juvenile and regenerating adult skin, we observed a small subset of dMCs (n>10) withdraw their long protrusions, round up their cell body, and rapidly extend microvilli reminiscent of the mature "mace-like" MC morphology (Figure 5F; Movies 6,7). I do not think movie 7 shows that. If it does, please indicate which of the cells shows this behavior.

****Optional:****

Published scRNASeq of the zebrafish skin exists and I am wondering if the authors could have searched for dMC and MC genes in these data which then could be used to generate lineage tracing tools or perform a pseudotime analysis that could indicate lineage relationships.

****Significance:****

The aim of the study was to test if motile, dividing dMCs are precursors of immotile, postmitotic MCs or a functionally distinct subpopulation of neuroendocrine cells. The manuscript is largely descriptive, well written and the findings are supported by beautiful imaging. The authors performed a series of experiments that strongly support the interpretation that dMCs are immature MCs. The findings will be of interest to developmental and stem cell biologists who study cell specification and differentiation. The most direct evidence that dMCs and MCs share a lineage relationship are the observations of a few dMCs that acquire the morphology of MCs in time lapse analyses. The other results support this interpretation but are correlative and do not exclude the possibility that dMCs are a functionally distinct cell type. To substantiate their interpretation the authors could take advantage of their photoconvertible line and photoconvert individual dMCs to determine if they differentiate into MCs.

Reviewer 2

Evidence, reproducibility and clarity

This work by Craig et al., defines intermediate steps in Merkel cell (MC) differentiation during development and regeneration in the zebrafish model system. Using live imaging, the authors describe a number of previously unappreciated steps that lead to the MCs differentiation from basal keratinocytes through a dendritic MC (dMC) intermediate. Live imaging of dMCs' microvilli as well as dMCs show a previously unrecognized dynamics of dMCs, including the presence of long actin-based protrusions and their dynamics. The authors also carefully analyzed dMCs migration, dynamics of dMC-dMC contacts and their division. Moreover, lineage tracing identified basal keratinocytes as dMC precursors, showing that basal keratinocytes give rise to this intermediate cell population. Their marker expression analysis provides further evidence that dMCs indeed represent a

transitional state between basal keratinocytes and MCs. They also look at the MCs renewal during skin regeneration and show that MCs in regenerated epidermis form predominantly de novo. Although the *Eda* requirement for MCs differentiation is not novel, they show that microvilli are absent in mutant cells. This adds some mechanistic insight into the MC protrusion formation. I found the study rigorous, well-controlled and their conclusions supported by the presented data. It clearly adds to our basic understanding of this important cell type. I only have a few general and minor comments.

****Major comments:****

One burning question is what controls the transition of dMCs into MCs? An obvious candidate is innervation. If the authors can demonstrate that, it would certainly take their work to another level.

What happens to the MC regeneration in *eda* mutants? Is it already known? If not, it would help to address its role in the MC differentiation process.

In their discussion they talk about directionality of MCs' protrusions in other species. Can they resolve MCs in 3D to address special orientation of their protrusions in zebrafish?

****Minor comments:****

The authors should comment on the *eda* expression; is it present in dMCs and MCs?

The difference between corneal and trunk dMCs and MCs in *eda* mutants is striking. The authors should comment on this in their discussion. Can they speculate on the basis of these differences?

****Referees cross-commenting****

Reviewer 3 made an important point about *atoh1a* expression and the reporter line. I agree that the authors should confirm their *atoh1a* reporter indeed marks dMCs and MCs.

****Significance:****

The strength of this work is the ability to follow MCs' differentiation in a live animal over time. One of its limitations is that the work is mostly descriptive. The main advance is showing that dMCs are the MCs intermediate population derived from basal keratinocytes. The study will be of an interest to sensory neuroscientists as well as those studying various aspects of skin development and regeneration.

Reviewer 3

Evidence, reproducibility and clarity

The manuscript from Craig et al., (2023) leverages a previously reported *atoh1a* reporter to drive expression of *lifeact-egfp* in Merkel cells (MC) to assess MC morphology during both scale development and regeneration, in the optically tractable zebrafish. Using a combination of live-imaging approaches and genetic perturbations, the authors show that MCs arise from a more immature population of dendritic Merkel cells (dMC) and that dMCs themselves derive from basal keratinocytes. The authors show that following injury, dMCs are the major cell type to infiltrate the regenerating scale region, with MCs becoming the predominant cell type at later stages of regeneration (presumably as the dMCs mature). The authors present evidence suggesting that dMCs are molecularly similar to both keratinocytes and MCs and argue that dMCs may represent an intermediate cell type. Data in the manuscript suggests MC and dMC protrusions are differently polarized, and that MC and dMC dynamics are also different. The authors provide direct evidence that dMCs mature into MCs morphologically and suggest that the reverse may also occur. Finally, the authors show that MC microvilli morphology is impaired in *eda*^{-/-} mutants, suggesting a role for *eda* in the normal morphology of MCs, more specifically in the trunk.

****Major comments:****

1. The discovery and characterization of dMCs in this study relies entirely on their labeling by an *atoh1a*-*lifeact* transgenic reporter. Given the striking similarity of dMCs to melanocytes, it is important to confirm the *atoh1a* reporter labels dMCs and MCs specifically, and not melanocytes. For example, it would be useful to see confirmation of cell type by double labelling of dMCs, e.g. with *atoh1a:lifeact-egfp* together with an antibody for *atoh1a* or preferably, another MC/dMC marker. dMCs look morphologically similar to melanocytes, which also display many of the behaviors noted in this manuscript. According to RNA-seq data (see <https://hair-gel.net/>), *atoh1* is

expressed in melanocytes in embryonic mouse skin and hair follicle stem cell precursors in post-natal skin. We recommend that the authors mine a similar dataset for zebrafish to ascertain whether *atho1a* is also expressed in pigment cells (e.g. <https://www.ncbi.nlm.nih.gov/geo/query/acc.cgi?acc=GSE190115>). We would also recommend that the authors run a stain for a melanocyte marker such as *Mitf/Tyr/Dct* to show this is not expressed in dMCs.

2. A major conclusion of the paper is that dMCs display molecular properties that overlap with both MCs and basal keratinocytes based on expression of three markers. I feel this conclusion is a little strong given the evidence presented; global transcriptomic analysis of these cells (RNA-seq) would better define where along a differentiation trajectory dMCs lie.

3. More data regarding the function of the dMC intermediate cell type would greatly strengthen the significance of the study. The characterization of dMCs forms the core of the report, yet little is shown/discussed regarding the function of this cell population. For example, why is this intermediary even required? Presumably this is to facilitate the migration of MCs from the basal layer into the upper strata and their dispersion upon arrival.

In this case, one could argue that the morphology of the dMC is directly related to its migratory function, as the authors suggest dMCs arise from basal keratinocytes, then migrate upwards towards the more superficial strata, where mature MCs are located.

However, very little evidence in support of this upward migration is presented - most of the migratory data are related to lateral movement. Experiments to alter the migratory properties of dMCs, for example using inhibitors of Arp2/3, would address whether migration is the key function of dMCs. Finally, there is insufficient evidence to suggest contact-inhibition is occurring, and in the cell division movie 5, it doesn't appear to happen (or the movie isn't long enough to show it). More examples are required or this observation should be reworded accordingly.

4. *Eda* is shown to be important for MC morphology, especially in MCs located in the trunk. More discussion of how *eda* may function would be helpful to the reader. For example, in what cells are *Eda* and *Edar* expressed? Do the authors think *Edar* signaling is cell autonomous within the MCs? Or does the loss of *Eda* indirectly affect MC morphology as a result of impaired scale formation? Additionally, the authors state that corneal MCs in both WT and *eda*^{-/-} have similar microvilli morphologies. The figure, however, shows that corneal MCs from these genotypes do look different, with *eda*^{-/-} corneal MCs having a more evenly distributed microvilli than the polarized microvilli of their WT counterparts. The metric '% of MCs with microvilli' does not capture this aspect of their morphology.

5. In several places, the number of biological replicates is unclear. A major concern is that data presented as 'number of cells' may only have been collated from n=1 animal. The authors should specify the number of both biological and technical replicates per experiment and consider displaying the data in superplots. Where stats are undertaken, particularly on percentages, it should be made clear whether the stats test was performed on raw numbers or the % (particularly true for Chi square). Examples of this issue can be found in figures 3CH, 4F-H, 5B-C and supplemental.

****Minor comments:****

- Line 124. Why did the authors choose developmental stages 11mm and 28mm for the quantification? The images in Figure 1 show 8, 10 and 12mm but not 11mm.

- Line 126. It is unclear what the difference is between circularity and roundness.

- Line 645 and Fig 11. 'Cells manually classified as MC or dMC'. Please provide further clarification on this categorization (e.g. number of protrusions/roundness value etc.)

- Line 141 and Fig 10. The authors comment on the mosaic nature of DsRed expression, but it seems particularly sparse in the image. Similarly, there are numerous GFP⁺ cells that do not express DsRed, and the ones that do are found at a distance from the DsRed⁺ basal keratinocytes. Further explanation is required here. For example, if MCs ultimately arise from dMCs, why are so few of them labelled? It would be useful to know the % of cre recombination that is actually occurring (i.e. how efficient the cre driver is in keratinocytes by DsRed⁺/total number) to put these data in context.

- Line 170 and 179. The authors do not comment on the possibility of de/trans-differentiation of mature MCs as an explanation of how dMCs and 'new' MCs arise on regenerating scales.

- Line 176. Can the authors comment on how quickly the nls-Eos protein turns over? This is pertinent given it is possible that by 7 dpp all the red nls-Eos could potentially have been replaced by green nls-Eos in an 'existing' *atho1a*⁺ cell.

- Figure 2M-P. Both channels (green and magenta) should be shown here. Cells will express both and it is unclear from the image panel what this looks like.
- Line 186, 200 and 206. 'regenerating dMCs' this is confusing. Perhaps reword to 'dMCs associated with regenerating scales'.
- Line 186. Why did the authors focus on 5dpp, particularly given that at 3 dpp the proportion of dMCs:MCs is more evenly spread?
- Figure 3A-B. An additional panel with DAPI is needed here to enable Tp63 negative nuclei to be visualized. Also, what is the cell in the top right of 3B? It has a red nucleus but is not marked by an asterisk.
- Figure 3D-E. This data panel also needs to show a dMC that is negative for SV2.
- Figure 4D-E and line 235. It is intuitive that dMCs will not have basal facing processes if they are already in the basal layer of keratinocytes - there simply isn't the physical space (unless they penetrate the scales/basement membrane which presumably they don't). Also, the authors need to comment on, and quantify dMC protrusions in relation to the directionality of dMC migration in the main text. This is referred to in line 762 as part of the figure legend (Fig 5) and Movie 3 legend (line 809), but this is not quantified anywhere.
- Line 258. How do these unipolar protrusions correlate with directionality?
- Line 287 and Figure 5G. There is insufficient evidence to conclude that MCs can revert back to dMCs, particularly given that MCs are considered post-mitotic. N=2 (cells/fish?) is not sufficient without further evidence, and the MC depicted in Figure 5G doesn't resemble a bona fide MC at the start of imaging. Suggest removing this conclusion and data or increasing n and providing further evidence.
- Line 394. 'These protrusions extended from lateral-facing membranes and interdigitated between basal and suprabasal keratinocytes'. Did the authors specifically show this? It is not clear from the data.
- Line 430. The reference to Merkel Cell carcinoma needs more commentary with regards to the relevance of the authors' findings.
- Line 491. Denoise.ai was used on images as stated. Can the authors confirm that any image quantification was done on raw images prior to using the Denoise.ai function?
- Line 528. Include details of the tp63 antibody here.

****Significance:****

Overall, the data are novel and of interest to researchers in several fields, including development, skin biology and MC carcinoma. This work provides an important step forward in our understanding of how basal keratinocytes give rise to MCs in zebrafish - via a dMC intermediary cell type. The imaging presented therein is of a high quality, and the movies are beautiful; capturing the cellular behaviors very clearly. This paper does not however, comment on the molecular mechanisms regulating this transition, nor on the cellular mechanisms resulting in the altered morphology and migration of dMCs and maturation into MCs. Inclusion of data as described above in the major comments section would increase the significance and impact of this work. Notwithstanding, the observations made in this work describe, for the first time to my knowledge, a morphologically distinct cell type in zebrafish (dMCs) similar to that having been described in other vertebrates and provide the ground work for future investigation.

Manuscript number: RC-2023-02186

Corresponding author(s): Jeffrey P. Rasmussen

1. General Statements [optional]

We would like to collectively thank the reviewers for their thoughtful feedback on our initial submission. We apologize for the lengthy revision process, but this was unavoidable due to lab personnel changes. Incorporating the reviewers' suggestions has helped us improve and strengthen the manuscript in several key areas. The main revisions to the manuscript include:

- Clarification on the specificity of the *atoh1a* reporter used through additional data (Figure S3; Author Response Figure 1) and revisions to the text.
- Addition of dMC innervation data as Figure S1.
- Analysis of SV2 staining intensity vs. cell morphology (Figure 3G).
- To better illustrate the complex 3D morphologies and positions of dMCs and MCs, we revised Figure 4 and consolidated the corresponding results text into a standalone

section. This includes the addition of two new microscopy panels (Figure 4E,F) and two new supplementary videos (Videos 1 and 2).

- We performed photoconversion of single dMCs and found that they show increases in circularity after 24 hrs, consistent with our DMC→MC maturation model. We incorporated this experiment and our live-imaging of maturation into to Figure 6 and as standalone results section.
- Further description of the *eda* mutant phenotype through addition of new data (exfoliation-induced regeneration) into the results section (Figure 8), *edar* HCR (Author Response Figure 2) and revisions to the discussion.
- Improved statistical reporting throughout.

We expand on the individual comments in our point-by-point response below (in blue font). Changes to the main text have been marked in blue font in the uploaded file.

Reviewer #1 (Evidence, reproducibility and clarity (Required)):

Summary:

In this manuscript, the authors use confocal imaging techniques to morphologically characterize Merkel cells during their maturation process in the zebrafish skin. Using an F-actin reporter, they identify two morphologically distinct populations of *atoh1a*⁺ cells: 1) Mature Merkel cells (MCs), which had previously been described in zebrafish, and 2) a transient population sharing morphological characteristics with so called dendritic Merkel Cells (dMCs), that were described in mice and humans but not previously identified in zebrafish. It was unknown whether dMCs represent a developmentally immature MC state or a functionally distinct subpopulation of neuroendocrine cells. The authors go on to show that dMCs represent the primary *atoh1a*⁺ cell type during skin regeneration and share features of both basal keratinocytes and Merkel cells, leading them to speculate that they could be MC precursors. Confocal time lapse imaging further showed that MCs and dMCs differ in the polarity of their protrusions. In some of the lapses, dMC can be seen maturing into MCs, providing evidence that they could be precursor cells. MC to dMC reversion events are also observed, albeit less often. Finally, the authors show that loss Ectodysplasin A (*Eda*) signaling disrupts MC microvilli formation, identifying this pathway as a potential regulator of MC morphology.

Major comments:

- The authors conclude that dMCs represent an intermediate state in the MC maturation program. This is based on the observation that the percentage of dMCs decreases over time and the fact that they share characteristics of both keratinocytes and MCs. In addition, dMCs are observed to mature into MCs in time lapses. However, these findings do not completely rule out the possibility that dMCs represent a transient, functionally distinct population of MCs. The authors should discuss this possibility.

Author response 1.1: We note that we establish that most dMCs express *piezo2* and SV2 (Figure 3E,H), consistent with a neurosecretory/mechanosensory profile of MCs. In the revision, we now include new data showing that ~90% of dMCs contact somatosensory axons (Figure S1), again consistent with MC function. Moreover, previous work showed that murine dMCs depolarize in response to mechanical stimulation (Maksimovic et al., 2014). Nevertheless, the reviewer correctly points out that we cannot fully exclude the possibility that “that dMCs represent a transient, functionally distinct population of MCs” based on our current data. In response to the reviewer’s comment, we added the following section to the discussion (L460-463) where we touch on these points (new additions in italics):

“... Consistent with this notion, we observed that most dMCs contacted axons and prior electrophysiological recordings from dMCs found that mechanical stimulation resulted in rapid depolarization of the cells (Maksimovic et al., 2014). Nevertheless, we cannot exclude the possibility of additional functional roles for dMCs.”

Additionally, some clarifications on the data could help strengthen their conclusion:

o Figure 1 I-K: The interpretation of the simultaneous increase of dMCs and MCs is not clear. Shouldn't the percent of dMCs be highest at 8-9mm and then go down, when MCs first start to appear?

Author response 1.2: This likely reflects the continuous addition of *atoh1a*⁺ cells to the epidermis throughout the 8-15 mm SL period (**Figure 1K**). There are few *atoh1a*⁺ cells at 8-9 mm and we have not focused on this stage. We revised the corresponding results section (L136-139) as follows: “Manual cell counting from our imaging dataset revealed dMCs appeared in highest frequency and density at 10-15 mm SL (**Figure 1I,J**). *During this period, scales expanded and total atoh1a+ cell density continuously increased (Figure 1K) consistent with our previous work (Brown et al., 2023).*”

o Fig. 2K: These results could also mean that dMCs numbers stay the same and only MCs increase in number. Does not imply lineage as stated in line 182 where the authors say that dMCs are a transient population. Please also report the total number of dMCs.

Author response 1.3: Thank you for the comment. In response to the reviewer's comment, we now plot the dMC density over the course of scale regeneration in **Figure 2L**. The revised results section reads as follows (L196-198): “After 3 dpp, the proportion *and density* of dMCs gradually decreased, with MCs becoming the predominant *atoh1a*⁺ cell type at later stages of regeneration (**Figure 2I,K,L**).” These observations are consistent with the “transient” interpretation.

We also updated the figure legend to include the total number of cells in the dataset as suggested: “*Total cells analyzed: 5764 MCs and 1064 dMCs.*”

o Figure 5 F and G: In these time lapses, “a small subset of dMCs (n>10)” is observed to adopt MC morphology. Does this mean 10 cells, and if so, out of how many? The authors should clarify how many time lapses were taken, and quantify the percentage of dMCs undergoing this process. The same goes for the reciprocal process, MC to dMC conversion, which happens only “in rare instances (n=2)”.

Author response 1.4: We agree with the reviewer that these are important details. We now include the details of our live-imaging dataset in **Table S1** and a summary in **Figure 4F**. Specifically, we now report: 1) the number of dMCs, MCs and fish imaged; 2) the length of each imaging session; and 3) the number of events (mitosis, maturation and cell death) observed. This allowed us to calculate the total “cell-hrs” in our dataset and the relative rate of events per cell-hrs. As suggested by Reviewer #3 (see author response 3.19), we removed our observations about the putative reversion events from the revision.

- Use photoconversion of single cells to establish lineage relationship. The 2 time lapses shown are not statistically significant and the identity of MCs in these movies is solely based on morphology.

Author response 1.5: Thank you for the suggestion. We first confirmed that we could use nuclear shape as a proxy for *atoh1a*⁺ cell morphology. To accomplish this we co-imaged *Tg(atoh1a:lifect-egfp)* with a new knock-in allele of *sox2*, which we found labels dMCs and MCs (**Figure S3**). We found a positive correlation between the nuclear and membrane circularity of *atoh1a*⁺ cells (**Figure S3E**), suggesting that nuclear morphology can serve as a proxy for the morphological cell state.

Next, as suggested by the reviewer, we performed single-cell photoconversion using *Tg(atoh1a:nls-eos)*. Consistent with our model, cells with ovular nuclei transitioned into cells with more spherical nuclei (**Figure 6A-C**). Our quantifications showed that cells that began with ovular nuclei significantly increased their nuclear circularity after 24 hrs of scale regeneration (**Figure 6D**). In our revised manuscript, we consolidated our time lapse results with these new photoconversion data into the results subsection “dMCs can directly mature into MCs” (L317-357) and the corresponding Figure 6. We also now present improved statistical reporting of our time-lapse data (**Figure 6F and**

Table S1).

- In the last part of the paper, the authors show that trunk dMCs and MCs adopt abnormal morphologies in the absence of Eda signaling. However, this phenotype is not seen in the corneal epidermis, which is not squamated. Since Eda mutants do not develop scales, could the altered morphology in the trunk be due to the absence of scales? If possible, the authors should inhibit Eda signaling after the formation of scales or tone down their conclusions.

Author response 1.6: Thank you for the comment. The reviewer is correct that we cannot currently distinguish between a direct effect of Eda signaling or a downstream (e.g., scale-dependent) signal without further genetic analysis. In response to the reviewer's comment, we revised the discussion (L517-519) to include the following point:

“Our study does not address whether these Eda-dependent phenotypes result from direct effects of Eda signaling on the MC lineage or from indirect effects, for example, through scale-derived signals.”

Please see also author response 2.4 regarding *edar* expression.

- Line 264: The authors write: 'Consistent with this notion, dMC-dMC or dMC-MC contacts resulted in lateral dMC movement away from the contact (Movie 4). Together these observations suggest that MCs are immotile, epithelial-like cells, whereas dMCs are motile, mesenchymal-like cells that undergo contact inhibition upon encountering another *atoh1a+* cell'. The lateral movement of dMCs after contacting MCs needs to be quantified before it can be interpreted as contact inhibition.

Author response 1.7: As we currently have limited examples of dMC-dMC and dMC-MC collisions we removed the comments about contact inhibition in the revision.

Minor comments:

- 'Defects in the morphogenesis of actin-based protrusions are linked to a variety of diseases, including colorectal cancer and deafness'. Please provide refs.

Author response 1.8: We apologize for this omission and thank the reviewer for the suggestion. We updated this sentence to include a reference to a recent review (Houdusse and Titus, 2021).

- Line 145: this experiment does not show motility. Just that basal keratinocytes give rise to them.

Author response 1.9: The reviewer correctly points out that other possibilities exist. We have deleted the motility statement from L145 and also revised L247 accordingly.

- Line 165. Cells increase by 14dpp and do not seem to plateau at 7dpp. Please discuss.

Author response 1.10: We revised this sentence as suggested: *“*atoh1a+* cells appeared above regenerating scales beginning at 2 dpp and increased in density until 14 dpp (Figure 2F-J).”*

- Line 190. Does Figure 3A not show basal keratinocytes? Only Figure 3B is cited.

Author response 1.11: **Figure 3A** shows suprabasal keratinocytes (now labeled in response to reviewer comment).

- Figure 3: Within individual cells, is there a negative correlation between SV2 staining and tp63 staining in dMCs? Or between sphericity and tp63 staining?

Author response 1.12: Thank you for the suggestion. In response to the reviewer's comment, we quantified the relationship between SV2 staining intensity and two metrics of cell morphology (circularity and perimeter), which revealed a positive

correlation These data are now presented in **Figure 3G**.

- If dMCs are immature, are they already innervated by somatosensory axons?

Author response 1.13: Thank you for raising this interesting question. To address the reviewer's question in trunk skin, we imaged animals co-expressing *Tg(ato1a:lifect-egfp)* and *Tg(p2rx3a:mCherry)*, a reporter for a subset of somatosensory neurons. We found that the majority of dMCs contacted *p2rx3a+* somatosensory axons and quantified whether these contacts were on the protrusions, cell body or both. This analysis has now been incorporated into the new **Figure S1** and the first results section (L149-156):

*"We previously found that most adult trunk MCs contact somatosensory axons (Brown et al., 2023). To determine whether dMCs also contacted somatosensory axons, we crossed *Tg(ato1a:lifect-egfp)* to *Tg(p2rx3a:mCherry)*, a reporter that labels a subset of cutaneous somatosensory axons (Palanca et al., 2013; Rasmussen et al., 2018), and acquired confocal z-stacks of juvenile skin from double transgenic fish (Figure S1A). We found that 90.5% of dMCs contacted *p2rx3a+* axons (n=38/42 cells; Figure S1A',B) and that axon contacts could occur at the cell body, on a protrusion, or both (Figure S1C).*

Thus, like MCs, dMCs frequently associate with cutaneous axons."

- Line 284: Indeed, during our live-imaging of juvenile and regenerating adult skin, we observed a small subset of dMCs (n>10) withdraw their long protrusions, round up their cell body, and rapidly extend microvilli reminiscent of the mature "mace-like" MC morphology (Figure 5F; Movies 6,7). I do not think movie 7 shows that. If it does, please indicate which of the cells shows this behavior.

Author response 1.14: In **Video 8** (referred to with the updated numbering), the cell above the label "dMC maturation" text label shows this behavior.

Optional:

Published scRNASeq of the zebrafish skin exists and I am wondering if the authors could have searched for dMC and MC genes in these data which then could be used to generate lineage tracing tools or perform a pseudotime analysis that could indicate lineage relationships.

Author response 1.15: Thank you for the suggestion. Based on our current and previous analyses that found the majority of MCs/dMCs along the trunk do not develop prior to squamation, we considered two published datasets that were generated from these later timepoints. The first was generated from dissected trunk skin at stage 9.6 SSL (Aman et al., 2023). The second was generated on FACS-purified *mitfa+* cells isolated from dissected, descaled adult skin (Frantz et al., 2023). Neither of these datasets were annotated by the authors as containing dMCs or MCs.

NOTE: We have removed unpublished data that had been provided for the referees in confidence.

By querying the Aman et al. (2023) data set with *ato1a* as a candidate MC-specific gene, we found no enrichment of *ato1a* expression in any of the cell clusters (**Author Response Figure 1**). *ato1a* was not present in the count matrix in the Frantz et al. dataset. These observations suggest that neither dataset captured dMCs or MCs. This is likely due to the first dataset being collected at a time when few dMCs/MCs are present in the skin and the second dataset involving a descaling step, which would be predicted to remove the MC-containing epidermis. We agree that generating a scRNA-seq dataset that captures dMCs and MCs will be a valuable resource. We are currently generating an appropriate dataset in house and these results will be forthcoming in the future.

Reviewer #1 (Significance (Required)):

The aim of the study was to test if motile, dividing dMCs are precursors of immotile, post-mitotic MCs or a functionally distinct subpopulation of neuroendocrine cells. The manuscript is

largely descriptive, well written and the findings are supported by beautiful imaging. The authors performed a series of experiments that strongly support the interpretation that dMCs are immature MCs. The findings will be of interest to developmental and stem cell biologists who study cell specification and differentiation. The most direct evidence that dMCs and MCs share a lineage relationship are the observations of a few dMCs that acquire the morphology of MCs in time lapse analyses. The other results support this interpretation but are correlative and do not exclude the possibility that dMCs are a functionally distinct cell type. To substantiate their interpretation the authors could take advantage of their photoconvertible line and photoconvert individual dMCs to determine if they differentiate into MCs.

We thank the reviewer for their positive assessment of our work and suggestion about the single-cell photoconversion experiment.

Reviewer #2 (Evidence, reproducibility and clarity (Required)):

This work by Craig et al., defines intermediate steps in Merkel cell (MC) differentiation during development and regeneration in the zebrafish model system. Using live imaging, the authors describe a number of previously unappreciated steps that lead to the MCs differentiation from basal keratinocytes through a dendritic MC (dMC) intermediate. Live imaging of dMCs' microvilli as well as dMCs show a previously unrecognized dynamics of dMCs, including the presence of long actin-based protrusions and their dynamics. The authors also carefully analyzed dMCs migration, dynamics of dMC-dMC contacts and their division. Moreover, lineage tracing identified basal keratinocytes as dMC precursors, showing that basal keratinocytes give rise to this intermediate cell population. Their marker expression analysis provides further evidence that dMCs indeed represent a transitional state between basal keratinocytes and MCs. They also look at the MCs renewal during skin regeneration and show that MCs in regenerated epidermis form predominantly de novo. Although the Eda requirement for MCs differentiation is not novel, they show that microvilli are absent in mutant cells. This adds some mechanistic insight into the MC protrusion formation. I found the study rigorous, well-controlled and their conclusions supported by the presented data. It clearly adds to our basic understanding of this important cell type. I only have a few general and minor comments.

We thank the reviewer for their positive feedback on our work.

Major comments:

One burning question is what controls the transition of dMCs into MCs? An obvious candidate is innervation. If the authors can demonstrate that, it would certainly take their work to another level.

Author response 2.1: We agree that this is an intriguing question. As discussed in author response 1.13, our revision now includes quantification of dMC innervation (**Figure S1**). While the hypothesis that innervation triggers the dMC→MC maturation is attractive, other possibilities exist. Testing whether innervation controls the dMC→MC transition will require mutant and experimental analysis that we respectfully suggest is beyond the scope of the current manuscript. In discussing the dMC→MC transition, we state (L498- 499): “Although we do not know yet what cues this maturation, it appears to involve a transition from a mesenchymal-like to an epithelial-like state.”

What happens to the MC regeneration in *eda* mutants? Is it already known? If not, it would help to address its role in the MC differentiation process.

Author response 2.2: To address this suggestion, we performed exfoliation (a form of mild skin injury) in *eda* mutants and sibling controls. This analysis is now presented in **Figure 8** and the corresponding results section (L386-403) as follows:

“We next questioned whether Eda regulated MC morphology during skin regeneration. As eda mutants lack scales, we sought an alternative injury model that would trigger MC regeneration. Mild injury can trigger MC regeneration in murine skin (Wright et al., 2017), and gentle exfoliation of zebrafish skin initiates superficial keratinocyte regeneration (Chen et al., 2016). Thus, we began by assessing MC regeneration

following exfoliation across a grid of adjacent scales with a cotton swab (Figure 8A). In controls, we found that skin exfoliation resulted in a near complete loss of superficial keratinocytes and *atoh1a*⁺ dMCs and MCs immediately following injury (Figure 8B-D,K,L). At 4 and 7 days post-exfoliation (dpe), we observed restratification of the epidermis and a reappearance of dMCs and MCs (Figure 8E,F,K,L). As expected, most regenerated MCs adopted morphologies similar to those seen in uninjured skin, including the presence of microvilli (Figure 8C,E,F,M). Similar to controls, exfoliation of the trunk of *eda*^{-/-} mutants resulted in a removal of epidermal layers that was followed by restratification and a reappearance of dMCs and MCs at 4 and 7 dpe (Figure 8G-L). However, similar to our observations in uninjured skin, a large proportion of the MCs that regenerated in *eda*^{-/-} mutant skin contained a ring-like localization of Lifeact-EGFP and lacked microvilli (Figure 8I,J,M). Thus, exfoliation in zebrafish provides a simple system to study MC regeneration independent of associated dermal appendage regeneration, and our results suggest that *Eda* promotes the normal morphology of MCs during epidermal regeneration.”

In their discussion they talk about directionality of MCs' protrusions in other species. Can they resolve MCs in 3D to address special orientation of their protrusions in zebrafish?

Author response 2.3: We apologize for not articulating our observations on MC protrusion polarity more clearly in our initial submission. We quantified MC protrusion polarity as part of original submission (original Fig 4F and G) based on reconstructed cross-sections from confocal z-stacks. In response to the reviewer's comment, we have modified Figure 4 to more clearly highlight our observations on dMC/MC protrusion orientation by adding representative images (Figure 4E,F) and reorganizing the figure.

To better illustrate the complex 3D geometry of *atoh1a*⁺ cells, we now also added new 3D rotations of an MC (Video 1) and 2 dMCs (Video 2) to the supplementary materials. Finally, we now reference these panels/videos in the corresponding section of the discussion.

Minor comments:

The authors should comment on the *eda* expression; is it present in dMSs and MCs?

Author response 2.4: Thank you for raising this question. Previous work found that *eda* is expressed in the dermis whereas *edar* is expressed in the epidermis during scale development (Aman et al., 2018). This is now mentioned in the results section (L362-364). To interrogate *edar* expression in relation to *Tg(atoh1a:lifeact-egfp)* expression, we stained regenerating scales with HCR probesets against *edar* at 5 dpp. The *edar* probe set was designed using *in_situ_generator* v.0.3.2 (Kuehn et al., 2022) (accession#: NM_001115064.2; set size: 23; amplifier B3) and imaged with a 40x oil immersion objective (N.A.1.3). Our staining for *edar* revealed signal predominantly in the basal epidermis that occasionally weakly overlapped with *atoh1a* expression (Author Response Figure 2).

NOTE: We have removed unpublished data that had been provided for the referees in confidence.

Based on these RNA localization results, it is possible that *Eda/Edar* signaling has an autonomous function within the MC lineage. The tools (e.g., floxed alleles) to genetically address lineage autonomy of *Eda/Edar* signaling do not currently exist in zebrafish, but this would certainly be an interesting avenue for future studies. Please see also author response 1.6 regarding updates to the discussion.

The difference between corneal and trunk dMCs and MCs in *eda* mutants is striking. The authors should comment on this in their discussion. Can they speculate on the basis of these differences?

Author response 2.5: We agree with the reviewer that this difference is intriguing. In response to the comment, we revised our discussion as follows: “Surprisingly, *whereas*

corneal MCs appeared unaffected, we found that most trunk MCs completely lacked microvilli in eda mutants, a novel MC phenotype to our knowledge. This observation indicates Eda functions upstream of a MC microvillar program within trunk skin, consistent with its known expression pattern (Aman et al., 2018; Harris et al., 2008)."

Referees cross-commenting

Reviewer 3 made an important point about *atoh1a* expression and the reporter line. I agree that the authors should confirm their *atoh1a* reporter indeed marks dMCs and MCs.

Author response 2.6: Thank you for the suggestion. We address this point in author response 3.1.

Reviewer #2 (Significance (Required)):

The strength of this work is the ability to follow MCs' differentiation in a live animal over time. One of its limitations is that the work is mostly descriptive. The main advance is showing that dMCs are the MCs intermediate population derived from basal keratinocytes. The study will be of an interest to sensory neuroscientists as well as those studying various aspects of skin development and regeneration.

Reviewer #3 (Evidence, reproducibility and clarity (Required)):

The manuscript from Craig et al., (2023) leverages a previously reported *atoh1a* reporter to drive expression of *lifeact-egfp* in Merkel cells (MC) to assess MC morphology during both scale development and regeneration, in the optically tractable zebrafish. Using a combination of live- imaging approaches and genetic perturbations, the authors show that MCs arise from a more immature population of dendritic Merkel cells (dMC) and that dMCs themselves derive from basal keratinocytes. The authors show that following injury, dMCs are the major cell type to infiltrate the regenerating scale region, with MCs becoming the predominant cell type at later stages of regeneration (presumably as the dMCs mature). The authors present evidence suggesting that dMCs are molecularly similar to both keratinocytes and MCs and argue that dMCs may represent an intermediate cell type. Data in the manuscript suggests MC and dMC protrusions are differently polarized, and that MC and dMC dynamics are also different. The authors provide direct evidence that dMCs mature into MCs morphologically and suggest that the reverse may also occur. Finally, the authors show that MC microvilli morphology is impaired in *eda*^{-/-} mutants, suggesting a role for *eda* in the normal morphology of MCs, more specifically in the trunk.

Major comments:

1. The discovery and characterization of dMCs in this study relies entirely on their labeling by an *atoh1a*-*lifeact* transgenic reporter. Given the striking similarity of dMCs to melanocytes, it is important to confirm the *atoh1a* reporter labels dMCs and MCs specifically, and not melanocytes. For example, it would be useful to see confirmation of cell type by double labelling of dMCs, e.g. with *atoh1a:lifeact-egfp* together with an antibody for *atoh1a* or preferably, another MC/dMC marker. dMCs look morphologically similar to melanocytes, which also display many of the behaviors noted in this manuscript. According to RNA-seq data (see <https://hair-gel.net/>), *atoh1* is expressed in melanocytes in embryonic mouse skin and hair follicle stem cell precursors in post-natal skin. We recommend that the authors mine a similar dataset for zebrafish to ascertain whether *atoh1a* is also expressed in pigment cells (e.g. <https://www.ncbi.nlm.nih.gov/geo/query/acc.cgi?acc=GSE190115>). We would also recommend that the authors run a stain for a melanocyte marker such as Mitf/Tyr/Dct to show this is not expressed in dMCs.

Author response 3.1: Thank you for raising these points. We agree with the reviewer that establishment of the dendritic *atoh1a*⁺ cell identity is important for our work.

Regarding confirmation of the dMC cell type by double labeling, we included data in our original submission to this effect. Specifically, we co-stained *Tg(atoh1a:lifeact-egfp)* with HCR probes against *piezo2* and an anti-SV2 antibody as part of Figure 3. In

both cases we found that a subset of dMCs were double-labeled. In the revised manuscript, we now further show co-expression of *atoh1a* and *sox2* reporters (**Figure S3**). We now state in the results section (L242-244): “In summary, dMCs display molecular properties that overlap with both basal keratinocytes and MCs, supporting the interpretation that dMCs represent a transitional or immature MC *rather than belonging to an alternative epidermal lineage.*”

Regarding the morphological and behavioral similarities of dMCs to melanocytes, this is an interesting point that we had not considered. While dMCs and melanocytes share some similarities, there are also differences, e.g.: 1) dMCs are not pigmented; 2) dMCs are much smaller than melanocytes; 3) most melanocytes in the regions of trunk skin we imaged are located in the lower layers of skin (hypodermis) than the superficially located dMCs (epidermis); and 4) melanocytes have not been reported to express SV2, *piezo2*, or *Sox2*. Nevertheless, as discussed above in author response 1.15, we analyzed the GSE190115 dataset from Frantz et al. (2023) and did not observe *atoh1a* expression in pigment cells in this dataset (**Author Response Figure 1**). We believe that this dataset is the most relevant since it contains pigment cells isolated from adult zebrafish skin.

2. A major conclusion of the paper is that dMCs display molecular properties that overlap with both MCs and basal keratinocytes based on expression of three markers. I feel this conclusion is a little strong given the evidence presented; global transcriptomic analysis of these cells (RNA-seq) would better define where along a differentiation trajectory dMCs lie.

Author response 3.2: This is a fair point. In response to the reviewer’s comments, we have toned down our conclusions and we now comment on this limitation in the discussion as follows (L455-457):

“We note that our current analysis is restricted to a limited set of molecular markers and that comparing the transcriptional profiles of keratinocytes, dMCs and MCs at genome- wide scale is a worthy future endeavor.”

3. More data regarding the function of the dMC intermediate cell type would greatly strengthen the significance of the study. The characterization of dMCs forms the core of the report, yet little is shown/discussed regarding the function of this cell population. For example, why is this intermediary even required? Presumably this is to facilitate the migration of MCs from the basal layer into the upper strata and their dispersion upon arrival. In this case, one could argue that the morphology of the dMC is directly related to its migratory function, as the authors suggest dMCs arise from basal keratinocytes, then migrate upwards towards the more superficial strata, where mature MCs are located. However, very little evidence in support of this upward migration is presented - most of the migratory data are related to lateral movement. Experiments to alter the migratory properties of dMCs, for example using inhibitors of Arp2/3, would address whether migration is the key function of dMCs. Finally, there is insufficient evidence to suggest contact- inhibition is occurring, and in the cell division movie 5, it doesn’t appear to happen (or the movie isn’t long enough to show it). More examples are required or this observation should be reworded accordingly.

Author response 3.3: Regarding the function of the dMC intermediates, we agree that dMCs likely exist to facilitate the positioning of MCs (both laterally and vertically within the epidermis). We revised the discussion (L492-494) to this effect: *“We speculate that a major function of the dMC intermediate is to promote proper positioning of MCs— both laterally and vertically—within the epidermis.”*

Regarding the comment about upward migration of dMCs, as mentioned in author response 2.3, we reorganized **Figure 4** and added **Videos 1 and 2** to better illustrate/convey the complex 3D geometry of MCs/dMCs. In particular, **Video 2** shows examples of dMCs with both laterally and vertically directed protrusions.

To attempt to analyze dMC cell motility in a pharmacologically accessible system, we explanted and imaged regenerating scales with time-lapse confocal microscopy at 5 and

7 dpp. We had hoped to treat these explants with the Arp2/3 inhibitor CK-666 as suggested by the reviewer, however, dMCs exhibited limited motility during these explant movies under control conditions. Thus, additional optimization, either in explanted scales or in vivo, will be required to appropriately address the cytoskeletal regulation of dMC behaviors.

As discussed in author response 1.7, we removed mention of contact inhibition from the revision.

4. Eda is shown to be important for MC morphology, especially in MCs located in the trunk. More discussion of how eda may function would be helpful to the reader. For example, in what cells are Eda and Edar expressed? Do the authors think Edar signaling is cell autonomous within the MCs? Or does the loss of Eda indirectly affect MC morphology as a result of impaired scale formation? Additionally, the authors state that corneal MCs in both WT and *eda*^{-/-} have similar microvilli morphologies. The figure, however, shows that corneal MCs from these genotypes do look different, with *eda*^{-/-} corneal MCs having a more evenly distributed microvilli than the polarized microvilli of their WT counterparts. The metric '% of MCs with microvilli' does not capture this aspect of their morphology.

Author response 3.4: Thank you for the suggestions. Regarding the expression patterns of *eda* and *edar*, please see author response 2.4. Regarding potential cell autonomous roles for Edar signaling, we did not attempt to address this question due to limitations of current tools and state this explicitly in the discussion (see also author responses 1.6 and 2.4).

We thank the reviewer for their astute observation about the corneal MC figure panels. Upon further investigation, the morphological difference was likely due to the original images inadvertently being from slightly different anatomical regions. In response to the reviewer's comment, we updated the panels with representative images that were reacquired from identical regions. We also revised the language in the main text to more cautiously interpret the corneal morphologies only in relation to the presence of microvilli in both genotypes (L377-378). We did not attempt to do more detailed quantifications of microvillar morphologies in the different skin compartments as this would require a different imaging modality, but we agree that this could be an interesting line of future research and that zebrafish would be well-suited to this type of study.

5. In several places, the number of biological replicates is unclear. A major concern is that data presented as 'number of cells' may only have been collated from n=1 animal. The authors should specify the number of both biological and technical replicates per experiment and consider displaying the data in superplots. Where stats are undertaken, particularly on percentages, it should be made clear whether the stats test was performed on raw numbers or the % (particularly true for Chi square). Examples of this issue can be found in figures 3C-H, 4F-H, 5B-C and supplemental.

Author response 3.5: Thank you for raising this point. We apologize for the lack of clarity on these details in our initial submission. In all cases, the data presented as 'number of cells' come from multiple biological replicates. We have clarified these details in the associated figure legends. We have also clarified that the statistical tests were performed on the raw numbers, not the percentages, in the "Statistical analysis" section of the methods.

Minor comments:

- Line 124. Why did the authors choose developmental stages 11mm and 28mm for the quantification? The images in Figure 1 show 8, 10 and 12mm but not 11mm.

Author response 3.6: 11 and 28 mm were chosen as two representative timepoints of juvenile and adult skin, respectively, to illustrate the increased variability of *atoh1a*⁺ cell shapes at juvenile stages. We have modified this line as follows: Next, we

quantified *atoh1a* cell shapes at *representative juvenile and adult stages (11 and 28 mm SL, respectively) ...*”

- Line 126. It is unclear what the difference is between circularity and roundness.

Author response 3.7: To clarify the measurements, we have added the formulae for circularity and roundness to the methods section (L417).

- Line 645 and Fig 11. 'Cells manually classified as MC or dMC'. Please provide further clarification on this categorization (e.g. number of protrusions/roundness value etc.)

Author response 3.8: In response to the reviewer's comment, we have added the following to the methods section (L708-710): "... to manually classify *atoh1a*+ cells as dMC (*oblong cell body and one or more filopodial-like protrusions*) or MC (*spherical cell body and microvilli-like protrusions*) based on cell morphology.

- Line 141 and Fig 10. The authors comment on the mosaic nature of DsRed expression, but it seems particularly sparse in the image. Similarly, there are numerous GFP+ cells that do not express DsRed, and the ones that do are found at a distance from the DsRed+ basal keratinocytes. Further explanation is required here. For example, if MCs ultimately arise from dMCs, why are so few of them labelled? It would be useful to know the % of cre-recombination that is actually occurring (i.e. how efficient the cre driver is in keratinocytes by DsRed+/total number) to put these data in context.

Author response 3.9: Thank you for the comment. We agree that the mosaic nature of the Cre tools are a limitation. We also note that this issue is not unique to the specific tools we used and likely reflects influence of neighboring genomic regions in the random integration methodology historically used in zebrafish (*tol2* transposase based) (Lalonde et al., 2022). Hopefully the next generation tools built on validated safe harbor landing sites will help overcome this limitation (Lalonde et al., 2024). Another potential confound is whether the "ubiquitous" promoter used (*actb2*) is truly ubiquitous.

As suggested by the reviewer, we quantified the recombination efficiency and revised the results section (L166-173) as follows: "Due to transgene mosaicism and/or incomplete Cre-ERT2 activation, not all basal keratinocytes expressed DsRed (**Figure 10**). Specifically, by thresholding the DsRed or BFP channel from maximum intensity projections through the trunk epidermis, we found that 39.6% of the epidermis was DsRed+, whereas 35.2% was BFP+ (71 scales from n=4 fish; overall recombination efficiency: 52.9%). Along with DsRed+ MCs (**Figure 10''**), we observed that a subset of dMCs in juvenile skin were DsRed+ (**Figure 10'-O**). Although only 14.5% (n=48/332) of dMCs expressed Tg(*actb2:LOXP-BFP-LOXP-DsRed*), of these 54.2% (n=26/48) were DsRed+, consistent with the overall rate of epidermal recombination."

- Line 170 and 179. The authors do not comment on the possibility of de/trans-differentiation of mature MCs as an explanation of how dMCs and 'new' MCs arise on regenerating scales.

Author response 3.10: The reviewer is correct that we cannot exclude this possibility based on our photoconversion experiment. In response to the reviewer's comment, we added the sentence (L201-203): "*de novo production may include differentiation from precursors and/or trans-differentiation of another cell type.*" and refer to the cells containing only non-photoconverted nls-Eos as "de novo produced" throughout the paragraph.

- Line 176. Can the authors comment on how quickly the nls-Eos protein turns over? This is pertinent given it is possible that by 7 dpp all the red nls-Eos could potentially have been replaced by green nls-Eos in an 'existing' *atoh1a*+ cell.

Author response 3.11: Thank you for raising this important point. In response to the reviewer's comments, we added the following sentence to the results section (L208-

209): “We previously established that photoconverted *nls-Eos* is stable in adult MCs for at least 1 month (Brown et al., 2023).”

- Figure 2M-P. Both channels (green and magenta) should be shown here. Cells will express both and it is unclear from the image panel what this looks like.

Author response 3.12: We updated these panels as suggested by the reviewer.

- Line 186, 200 and 206. 'regenerating dMCs' this is confusing. Perhaps reword to 'dMCs associated with regenerating scales'.

Author response 3.13: Thank you for the suggestion. We revised these sentences as suggested.

- Line 186. Why did the authors focus on 5dpp, particularly given that at 3 dpp the proportion of dMCs:MCs is more evenly spread?

Author response 3.14: We focused on this timepoint for technical reasons: 3 dpp scales are quite small/flimsy and do not hold up well in our immunostaining protocol. We have clarified this point in the revised result section (L219-220) as follows: “...we focused on 5 dpp, a timepoint *at which scales are large enough to withstand immunostaining and contain a mix of dMCs and MCs (Figure 2K).*”

- Figure 3A-B. An additional panel with DAPI is needed here to enable Tp63 negative nuclei to be visualized. Also, what is the cell in the top right of 3B? It has a red nucleus but is not marked by an asterisk.

Author response 3.15: We added in the DAPI channel and better annotated the revised figure panels as suggested.

- Figure 3D-E. This data panel also needs to show a dMC that is negative for SV2.

Author response 3.16: Revised as suggested.

- Figure 4D-E and line 235. It is intuitive that dMCs will not have basal facing processes if they are already in the basal layer of keratinocytes - there simply isn't the physical space (unless they penetrate the scales/basement membrane which presumably they don't). Also, the authors need to comment on, and quantify dMC protrusions in relation to the directionality of dMC migration in the main text. This is referred to in line 762 as part of the figure legend (Fig 5) and Movie 3 legend (line 809), but this is not quantified anywhere.

Author response 3.17: As now better illustrated in the revised **Figure 4** and **Video 2**, dMCs are positioned above the basal keratinocyte layer. So it is possible for them to form basal facing processes without penetrating the scale/basement membrane. We apologize for the confusion and in response to the reviewer's comment, we revised the results section for **Figure 4** to better articulate the 3D anatomy.

Regarding the comment about dMC migration in relation to protrusion orientation, please see author response 3.18.

- Line 258. How do these unipolar protrusions correlate with directionality?

Author response 3.18: In response to the reviewer comment, we clarified this the text (L295) as follows: “dMCs were most motile when adopting elongated, ovoid cell bodies with long, unipolar protrusions at one end of the cell, *which correlated with the direction of migration in 95% of cells (n=20/21 from 9 fish)*”

- Line 287 and Figure 5G. There is insufficient evidence to conclude that MCs can revert back to dMCs, particularly given that MCs are considered post-mitotic. N=2 (cells/fish?) is not sufficient without further evidence, and the MC depicted in Figure 5G doesn't resemble a bona fide MC at

the start of imaging. Suggest removing this conclusion and data or increasing n and providing further evidence.

Author response 3.19: We did not mean to suggest that these were fully mature MCs, rather that *atoh1a*⁺ cells with a spherical, MC-like morphology can transition into an ovoid, dMC-like morphology. Nevertheless, in light of the reviewer's comment and since we currently do not have additional examples of this behavior, we removed the reversion data from the revised manuscript.

- Line 394. 'These protrusions extended from lateral-facing membranes and interdigitated between basal and suprabasal keratinocytes'. Did the authors specifically show this? It is not clear from the data.

Author response 3.20: As mentioned in author responses 2.3 and 3.3, to better illustrate the dMC protrusion morphologies and positioning within the epidermis, we now include higher-resolution views of dMCs in relation to the epidermal anatomy (Figure 4F and Video 2).

- Line 430. The reference to Merkel Cell carcinoma needs more commentary with regards to the relevance of the authors' findings.

Author response 3.21: We agree that this needed clarification. We have revised this section as follows (L520-525): "...Our observations that dMCs can divide suggests that they have not yet exited the cell cycle and, thus, could potentially serve as the cell type of origin for Merkel cell carcinoma. Moreover, the mesenchymal-like behaviors of dMCs are reminiscent of metastatic cells. Consistent with these ideas, recent transcriptional analyses identified gene expression associated with a partial mesenchymal state in human Merkel cell carcinoma tumor samples (Das et al., 2023; Karpinski et al., 2023)."

- Line 491. Denoise.ai was used on images as stated. Can the authors confirm that any image quantification was done on raw images prior to using the Denoise.ai function?

Author response 3.22: We recognize the reviewer's concern. We did perform the image quantification on images processed with Denoise.ai as stated in the methods based on recommendations from the manufacturer that these images remain quantitative (see: <https://www.nature.com/articles/d42473-019-00355-6>).

- Line 528. Include details of the tp63 antibody here.

Author response 3.23: Thank you for pointing out this missing detail. Corrected.

Reviewer #3 (Significance (Required)):

Overall, the data are novel and of interest to researchers in several fields, including development, skin biology and MC carcinoma. This work provides an important step forward in our understanding of how basal keratinocytes give rise to MCs in zebrafish - via a dMC intermediary cell type. The imaging presented therein is of a high quality, and the movies are beautiful; capturing the cellular behaviors very clearly. This paper does not however, comment on the molecular mechanisms regulating this transition, nor on the cellular mechanisms resulting in the altered morphology and migration of dMCs and maturation into MCs. Inclusion of data as described above in the major comments section would increase the significance and impact of this work. Notwithstanding, the observations made in this work describe, for the first time to my knowledge, a morphologically distinct cell type in zebrafish (dMCs) similar to that having been described in other vertebrates and provide the ground work for future investigation.

Reviewer expertise: skin biology, live-imaging, zebrafish, mouse, developmental biology.

We thank the reviewer for stressing the novelty of our work.

Author Response References

- Aman, A.J., A.N. Fulbright, and D.M. Parichy. 2018. Wnt/ β -catenin regulates an ancient signaling network during zebrafish scale development. *eLife*. 7. doi:10.7554/eLife.37001.
- Aman, A.J., L.M. Saunders, A.A. Carr, S. Srivatasan, C. Eberhard, B. Carrington, D. Watkins-Chow, W.J. Pavan, C. Trapnell, and D.M. Parichy. 2023. Transcriptomic profiling of tissue environments critical for post-embryonic patterning and morphogenesis of zebrafish skin. *eLife*. 12:RP86670. doi:10.7554/eLife.86670.
- Chen, C.-H., A. Puliafito, B.D. Cox, L. Primo, Y. Fang, S. Di Talia, and K.D. Poss. 2016. Multicolor cell barcoding technology for long-term surveillance of epithelial regeneration in zebrafish. *Dev. Cell*. 36:668-80. doi:10.1016/j.devcel.2016.02.017.
- Frantz, W.T., S. Iyengar, J. Neiswender, A. Cousineau, R. Maehr, and C.J. Ceol. 2023. Pigment cell progenitor heterogeneity and reiteration of developmental signaling underlie melanocyte regeneration in zebrafish. *eLife*. 12:e78942. doi:10.7554/eLife.78942.
- Houdusse, A., and M.A. Titus. 2021. The many roles of myosins in filopodia, microvilli and stereocilia. *Curr. Biol. CB*. 31:R586-R602. doi:10.1016/j.cub.2021.04.005.
- Kuehn, E., D.S. Clausen, R.W. Null, B.M. Metzger, A.D. Willis, and B.D. Özpolat. 2022. Segment number threshold determines juvenile onset of germline cluster expansion in *Platynereis dumerilii*. *J. Exp. Zool. B Mol. Dev. Evol.* 338:225-240. doi:10.1002/jez.b.23100.
- Lalonde, R.L., C.L. Kemmler, F.W. Riemsdagh, A.J. Aman, J. Kresoja-Rakic, H.R. Moran, S. Nieuwenhuize, D.M. Parichy, A. Burger, and C. Mosimann. 2022. Heterogeneity and genomic loci of ubiquitous transgenic Cre reporter lines in zebrafish. *Dev. Dyn. Off. Publ. Am. Assoc. Anat.* 251:1754-1773. doi:10.1002/dvdy.499.
- Lalonde, R.L., H.H. Wells, C.L. Kemmler, S. Nieuwenhuize, R. Lerma, A. Burger, and C. Mosimann. 2024. pIGLET: Safe harbor landing sites for reproducible and efficient transgenesis in zebrafish. *Sci. Adv.* 10:eadn6603. doi:10.1126/sciadv.adn6603.
- Maksimovic, S., M. Nakatani, Y. Baba, A.M. Nelson, K.L. Marshall, S.A. Wellnitz, P. Firozi, S.-H. Woo, S. Ranade, A. Patapoutian, and E.A. Lumpkin. 2014. Epidermal Merkel cells are mechanosensory cells that tune mammalian touch receptors. *Nature*. 509:617-21. doi:10.1038/nature13250.
- Wright, M.C., G.J. Logan, A.M. Bolock, A.C. Kubicki, J.A. Hemphill, T.A. Sanders, and S.M. Maricich. 2017. Merkel cells are long-lived cells whose production is stimulated by skin injury. *Dev. Biol.* 422:4-13. doi:10.1016/j.ydbio.2016.12.020.

Original submission

First decision letter

MS ID#: dev.204810

MS TITLE: Dendritic atoh1a+ cells serve as Merkel cell precursors during skin development and regeneration

AUTHORS: Evan W. Craig, Erik C. Black, Samantha Z. Fernandes, Ahlan S. Ferdous, Camille E. A. Goo, Sheridan Sargent, Elgene J. A. Quitevis, Avery Angell Swearer, Nathaniel G. Yee, Jimann Shin, Lilianna Solnica-Krezel and Jeffrey P. Rasmussen

Dear Jeff:

I am happy to tell you that all 3 reviewers were enthusiastic about your submission, and your manuscript has been accepted for publication in *Development*, pending our standard publication integrity checks.

Thank you for sending your manuscript to Development through Review Commons

Reviewer 1: This is a revised manuscript by Craig et al. that focuses on Merkel cell (MC) precursors in the zebrafish model. A previous version of the paper already did a good job characterizing this precursor population and defining its properties. I had three major comments concerning the role of innervation during the transition of dMCs into MCs, MC regeneration in *eda* mutants, and the spatial orientation of MC protrusions. I also agreed with another reviewer's comment about the need to clarify expression from the *atoh1a*-driven transgene, as it is a key reagent in this work.

In the revised manuscript, the authors have adequately addressed my concerns. They investigated MC regeneration in *eda* mutants, clarified the issue regarding MC protrusions, and provided additional data on MC innervation. Based on their original and new data, they convincingly argue that the *atoh1a:lifect-eGFP* transgene is indeed expressed in MCs. Finally, they also addressed all of my minor comments.

In summary, this is a solid study addressing an important developmental question: the differentiation of MCs during development and regeneration.

Reviewer 2: COMMENTS ON TEXT

The revision made the paper a lot stronger. The authors addressed all of my comments either experimentally or by toning down their conclusions. I think the innervation data is very nice and the comments I had about their time lapses were addressed sufficiently by providing more details on the imaging.

Reviewer 3: This manuscript describes the development and lineage of a Merkel Cell precursor population in the zebrafish skin (dMCs). The precursors are dendritic in morphology, arise from basal keratinocytes and share properties of both Merkel Cells and basal cells. Through live imaging, lineage tracing and photoconversion experiments, the authors convincingly show that dMCs mature into MCs during skin development and regeneration, and that the EDAR pathway is required for their mature morphology. The authors have responded to most of the reviewer comments and concerns from the Review Commons through extensive new data acquisition and quantification that bolster the conclusions that dMCs represent an MC precursor population. The study lays the groundwork for future studies interrogating the function, differentiation and innervation of this important sensory cell type and will be of interest to those studying epidermal development, regeneration, and mechanosensation.

COMMENTS ON DISPLAY ITEMS

Data are high quality, convincing and displayed in a well-organized manner.